# A multi-institutional study using artificial intelligence to provide reliable and fair feedback to surgeons

Dani Kiyasseh [1✉], Jasper Laca[2], Taseen F. Haque [2], Brian J. Miles[3], Christian Wagner[4], Daniel A. Donoho[5], Animashree Anandkumar[1] & Andrew J. Hung [2✉]

## Abstract

**Background** Surgeons who receive reliable feedback on their performance quickly master the skills necessary for surgery. Such performance-based feedback can be provided by a recently-developed artificial intelligence (AI) system that assesses a surgeon's skills based on a surgical video while simultaneously highlighting aspects of the video most pertinent to the assessment. However, it remains an open question whether these highlights, or explanations, are equally reliable for all surgeons.

**Methods** Here, we systematically quantify the reliability of AI-based explanations on surgical videos from three hospitals across two continents by comparing them to explanations generated by humans experts. To improve the reliability of AI-based explanations, we propose the strategy of training with explanations –TWIX –which uses human explanations as supervision to explicitly teach an AI system to highlight important video frames.

**Results** We show that while AI-based explanations often align with human explanations, they are not equally reliable for different sub-cohorts of surgeons (e.g., novices vs. experts), a phenomenon we refer to as an explanation bias. We also show that TWIX enhances the reliability of AI-based explanations, mitigates the explanation bias, and improves the performance of AI systems across hospitals. These findings extend to a training environment where medical students can be provided with feedback today.

**Conclusions** Our study informs the impending implementation of AI-augmented surgical training and surgeon credentialing programs, and contributes to the safe and fair democratization of surgery.

## Plain language summary

Surgeons aim to master skills necessary for surgery. One such skill is suturing which involves connecting objects together through a series of stitches. Mastering these surgical skills can be improved by providing surgeons with feedback on the quality of their performance. However, such feedback is often absent from surgical practice. Although performance-based feedback can be provided, in theory, by recently-developed artificial intelligence (AI) systems that use a computational model to assess a surgeon's skill, the reliability of this feedback remains unknown. Here, we compare AI-based feedback to that provided by human experts and demonstrate that they often overlap with one another. We also show that explicitly teaching an AI system to align with human feedback further improves the reliability of AI-based feedback on new videos of surgery. Our findings outline the potential of AI systems to support the training of surgeons by providing feedback that is reliable and focused on a particular skill, and guide programs that give surgeons qualifications by complementing skill assessments with explanations that increase the trustworthiness of such assessments.

[1]Department of Computing and Mathematical Sciences, California Institute of Technology, Pasadena, CA, USA. [2]Center for Robotic Simulation and Education, Catherine & Joseph Aresty Department of Urology, University of Southern California, Los Angeles, CA, USA. [3]Department of Urology, Houston Methodist Hospital, Houston, TX, USA. [4]Department of Urology, Pediatric Urology and Uro-Oncology, Prostate Center Northwest, St. Antonius-Hospital, Gronau, Germany. [5]Division of Neurosurgery, Center for Neuroscience, Children's National Hospital, Washington DC, USA. ✉email: danikiy@hotmail.com; ajhung@gmail.com

Surgeons seldom receive feedback on how well they perform surgery[1,2], despite evidence that it accelerates their acquisition of skills (e.g., suturing)[3–7]. Such feedback can be automated, in theory, by artificial intelligence systems[8,9]. To that end, we recently developed a surgical artificial intelligence system (SAIS) that assesses the skill of a surgeon based on a video of intraoperative activity and simultaneously highlights video frames of pertinent activity. We demonstrated that SAIS reliably automates surgeon skill assessment[10] and also examined the fairness of its assessments[11]. In pursuit of developing a trustworthy AI system capable of safely automating the provision of feedback to surgeons, we must ensure that SAIS' video-frame highlights, which we refer to as AI-based explanations, align with the expectations of expert surgeons (i.e., be reliable)[12,13] and be equally reliable for all surgeons (i.e., be fair)[14]. However, it remains an open question whether AI-based explanations are reliable and fair. If left unchecked, misguided feedback can hinder the professional development of surgeons and unethically disadvantage one surgeon sub-cohort over another (e.g., novices vs. experts).

Although an explanation can take on many forms, it often manifests as the relative importance of data (attention scores) in disciplines where the attention-based transformer architecture[15] is used, such as natural language processing[16] and protein modelling[17,18]. An element with a higher attention score is assumed to be more important than that with a lower score. Similarly, with a transformer architecture which operates on videos, SAIS[10] generates an attention score for each frame in a surgical video, with high-attention frames assumed to be more relevant to the surgeon skill assessment. When such an AI system's explanations align with those provided by humans, it can direct surgeons to specific aspects of their operating technique that require improvement while simultaneously enhancing its trustworthiness[19]. As such, there is a pressing need to quantify the reliability of explanations generated by surgical AI systems.

Previous studies have investigated the reliability of AI-based explanations that highlight, for example, important patches of a medical image[20–22] or clinical variables[23]. However, these studies remain qualitative and thereby do not systematically investigate whether explanations are consistently reliable across data points. Studies that quantitatively evaluate AI-based explanations often exclude a comparison to human explanations[24,25], a drawback that extends to the preliminary studies aimed at also assessing the fairness of such explanations[26,27]. Notably, previous work has not quantitatively compared AI-based explanations to human explanations in the context of surgical videos, nor has it proposed a strategy to enhance the reliability and fairness of such explanations.

In this study, we quantify the reliability and fairness of explanations generated by a surgical AI system –SAIS[10] –that we previously developed and which was shown to reliably assess the skill level of surgeons from videos. Through evaluation on data from three geographically-diverse hospitals, we show that SAIS' attention-based explanations often align, albeit imperfectly, with human explanations. We also find that SAIS generates different quality explanations for different surgeon sub-cohorts (e.g., novices vs. experts), which we refer to as an explanation bias. To address this misalignment between AI-based and human explanations, we devise a general strategy of training with explanations –TWIX –which uses human explanations as supervision to explicitly teach an AI system to highlight important video frames. We find that TWIX enhances the reliability of AI-based explanations, mitigates the explanation bias, and improves the performance of skill assessment systems across hospitals. With SAIS likely to provide feedback to medical students in the near future, we show that our findings extend to a training environment. Our study suggests that SAIS, when used alongside TWIX, has the potential to provide surgeons with accurate feedback on how to improve their operating technique.

## Methods

**Surgical video samples**. In a previous study, we trained and deployed an AI system (SAIS) on videos of a surgical procedure known as a robot-assisted radical prostatectomy (RARP). The purpose of this procedure is to treat cancer by removing a cancerous prostate gland from the body of a patient. In general, to complete a surgical procedure, a surgeon must often perform a sequence of steps. We focus on one particular step of the RARP procedure, referred to as vesico-urethral anastomosis (VUA), in which the bladder and the urethra are connected to one another through a series of stitches. To perform a single stitch, a surgeon must first grasp the needle with a robotic arm (needle handling), and push that needle through tissue (needle driving), before following through and withdrawing the needle from the tissue to prepare for the next stitch (needle withdrawal). In this study, we leveraged SAIS' ability to assess the skill-level of needle handling and needle driving when provided with a video sample depicting these activities.

*Live surgical procedure.* To obtain video samples from videos of surgical procedures, we adopted the following strategy. Given a video of the VUA step ($\approx 20$ min), we first identified the start and end time of each stitch (up to 24 stitches) involved in completing this step. We subsequently identified the start and end time of the needle handling and needle driving activities performed as part of each stitch. A single video sample reflects one such needle handling (or needle driving) activity ($\approx 20 - 30$s). As such, each VUA step may result in approximately 24 video samples for each activity (see Table 1 for total number of video samples).

*Training environment.* We also designed a realistic training environment in which medical students, without any prior robotic experience, sutured a gel-like model of the bladder and urethra with a robot that is used by surgeons to perform live surgical procedures. To control for the number of stitches performed by each participant, we marked the gel-like model with 16 target entry/exit points, resulting in 16 video samples from each participant. Although each stitch involved a needle handling and needle driving activity (among others), we only focused on needle handling for the purpose of our analysis (see Table 1 for number of video samples).

**Ethics approval**. All datasets (data from University of Southern California, St. Antonius Hospital, and Houston Methodist Hospital) were collected under Institutional Review Board (IRB) approval from the University of Southern California in which informed consent was obtained (HS-17-00113). We have also obtained informed consent to present surgical video frames in figures.

**Skill assessment annotations**. We assembled a team of trained human raters to annotate video samples with skill assessments based on a previously-developed skill assessment taxonomy (also known as an end-to-end assessment of suturing expertise or EASE[28]). EASE was formulated through a rigorous Delphi process involving five expert surgeons that identified a strict set of criteria for assessing multiple skills related to suturing (e.g., needle handling, needle driving, etc.). Our team of raters comprised medical students and surgical residents who either helped devise the original skill assessment taxonomy or had been intimately aware of its details.

**Table 1 Total number of videos and video samples associated with each of the hospitals and tasks.**

| Task | Activity | Details | Hospital | Videos | Video samples | Surgeons | Generalizing to |
|---|---|---|---|---|---|---|---|
| skill assessment | suturing | needle handling | **USC** | 78 | 912 | 19 | videos |
| | | | SAH | 60 | 240 | 18 | hospitals |
| | | | HMH | 20 | 184 | 5 | hospitals |
| | | | LAB | 69 | 328 | 38 | modality |
| | | needle driving | **USC** | 78 | 530 | 19 | videos |
| | | | SAH | 60 | 280 | 18 | hospitals |
| | | | HMH | 20 | 220 | 5 | hospitals |

Note that we train our model, SAIS, on data exclusively shown in bold following a 10-fold Monte Carlo cross-validation setup. For an exact breakdown of the number of video samples in each fold and training, validation, and test split, please refer to Supplementary Tables 2–7. The data from the remaining hospitals are exclusively used for inference. SAIS is always trained and evaluated on a class-balanced set of data whereby each category (e.g., low skill and high skill) contains the same number of samples. This prevents SAIS from being negatively affected by a sampling bias during training, and allows for a more intuitive appreciation of the evaluation results. USC University of Southern California, SAH St. Antonius Hospital, HMH Houston Methodist Hospital.

*Exact criteria for skill assessment annotations.* The skill-level of needle handling is assessed by observing the number of times a surgeon had to reposition their grasp of the needle. Fewer repositions imply a high skill-level, as it is indicative of improved surgeon dexterity and intent. The skill-level of needle driving is assessed by observing the smoothness with which a surgeon pushes the needle through tissue. Smoother driving implies a high skill-level, as it is less likely to cause physical trauma to the tissue.

*Mitigating noise in skill assessment annotations.* We took several steps to mitigate the degree of noise in the skill assessment annotations. First, EASE outlines a strict set of criteria related to the visual and motion content of a video sample, thereby making it straightforward to identify whether such criteria are satisfied (or violated) upon watching a video sample. This reduces the level of expertise that a rater must ordinarily have in order to annotate a video sample. Second, the raters involved in the annotation process were either a part of the development of the EASE taxonomy or intimately aware of its details. This implied that they were comfortable with the criteria outlined in EASE. Third, and understanding that raters can be imperfect, we subjected them to a training process whereby raters were provided with a training set of video samples and asked to annotate them independently of one another. This process continued until the agreement of their annotations, which was quantified via inter-rater reliability, exceeded 80%. We chose this threshold based on (a) the level of agreement first reported in the study developing the EASE taxonomy and (b) an appreciation that natural variability is likely to exist from one rater to the next in, for example, the amount of attention they place on certain content within a video sample.

After completing their training process, raters were asked to annotate the video samples in this study with binary skill assessments (low vs. high skill). In the event of disagreements in the annotations, we followed the same strategy adopted in the original study[10] where the lowest of all scores is considered as the final annotation.

**Skill explanation annotations**. We assembled a team of two trained human raters to annotate each video sample with segments of time (or equivalently, spans of frames) deemed relevant for a particular skill assessment. We define segments of time as relevant if they reflect the strict set of criteria (or their violation) outlined in the EASE skill assessment taxonomy[28]. In practice, we asked raters to exclusively annotate video samples tagged as low skill from a previous study (for motivation, see later section). To recap, for the activity of needle handling, a low skill assessment is characterized by three or more grasps of the needle by the surgical instrument. For the activity of needle driving, a low skill assessment is characterized by either four or more adjustments of the needle when being driven through tissue or its complete removal from tissue in the opposite direction to which it was inserted. As such, raters had to identify both visual and motion cues in the surgical field of view in order to annotate segments of time as relevant. We reflect important time segments with a value of 1 and all other time segments with a value of 0.

*Visualising explanation annotations.* Consider that for a given video sample 30 s in duration, a human rater might annotate the first five seconds (0−5 s) as most important for the skill assessment. Based on our annotations, we show that ≈ 30% of a single video sample is often identified as important (see Supplementary Table 1). To visualise these annotations, we considered unique video samples in the test set of each of the 10 Monte Carlo folds. Since each video sample may vary in duration, and to facilitate a comparison of the heatmaps across hospitals, we first normalized the time index of each explanation annotation such that it ranged from 0 (beginning of video sample) to 1 (end of video sample). For needle handling, this translates to the beginning and end of needle handling, respectively. A value of 0.20 would therefore refer to the first 20% of the video sample. We subsequently averaged the explanation annotations, whose values are either 0 (irrelevant frame) or 1 (relevant frame), across the video samples for this normalized time index. We repeated the process for all hospitals and the skills of needle handling and needle driving (see Fig. 1).

*Training the raters.* Before providing such explanation annotations, however, the raters underwent a training process akin to the one conducted for skill assessment annotations. First, raters were familiarized with the criteria outlined in the skill assessment taxonomy. In practice, and to mitigate potential noise in the explanation annotations, our assembled team of raters had, in the past, been involved in providing skill assessment annotations while using the same exact taxonomy. The raters were then provided with a training set of low-skill video samples and asked to independently annotate them with segments of time that they believed were important to that skill assessment. During this time, raters were encouraged to abide by the strict set of criteria outlined in the skill assessment taxonomy. This training process continued until the agreement in their annotations, which was quantified via the intersection over union, exceeded 0.80. This implies that, on average, each segment of time highlighted by one rater exhibited an 80% overlap with that provided by another rater. This value was chosen, as with the skill assessment annotation process, with the appreciation that natural variability in the annotation process is likely to occur. Raters may disagree, for example, on when an important segment of time ends even when both of their explanation annotations capture the bulk of the relevant activity.

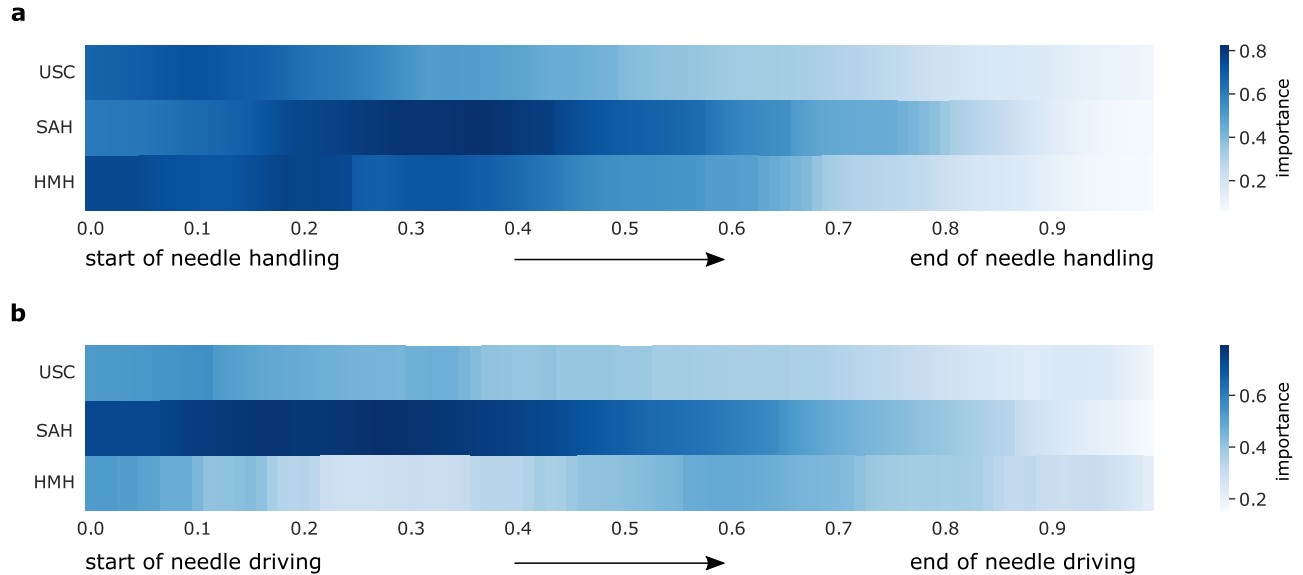

**Fig. 1 Heatmap of the ground-truth explanation annotations across hospitals.** We average the explanation annotations for the **a**, needle handling and **b**, needle driving video samples in the test set of the Monte Carlo folds (see Supplementary Table 2 for total number of samples), and present them over a normalized time index, where 0 and 1 reflect the beginning and end of a video sample, respectively. A darker shade (which ranges from 0 to 1 as per the colour bars) implies that a segment of time is of greater importance.

*Aggregating explanation annotations.* Upon completing the training process, raters were asked to provide explanation annotations for the video samples used in this study. They were informed that each video sample had been annotated in the past as low skill, and were therefore aware of the specific criteria in the taxonomy to look out for. In the event of disagreements in the explanation annotations, we considered the intersection of the annotations. This ensures that we avoid identifying potentially superfluous video frames as relevant and makes us more confident in the segments of time that overlapped amongst the raters' annotations. Although we experimented with other strategies for aggregating the explanation annotations, such as considering their union, we found this to have a minimal effect on our findings.

*Motivation behind focusing on low-skill activity.* In this study, our goal was to provide feedback for video samples exclusively depicting low skill activity. A binary skill assessment system is therefore well aligned with this goal. We focused on low-skill activity for two reasons. First, from a practical standpoint, it is relatively more straightforward to provide an explanation annotation for a video sample depicting low skill activity than it is for one depicting high skill activity. This is because human raters simply have to look for segments of time in the video sample during which one (or more) of the criteria outlined in EASE are violated. Second, from an educational standpoint, studies in the domain of educational psychology have demonstrated that corrective feedback following an error is instrumental to learning [1]. As such, our focus on a low skill activity (akin to an error) provides a ripe opportunity for the provision of feedback. We do appreciate, however, that feedback can also be useful when provided for video samples depicting high skill activity (e.g., through positive reinforcement). We leave this as an extension of our work for the future.

**Metrics for evaluating the reliability of explanations.** To evaluate the reliability of AI-based explanations, we compared them to human-based ground-truth explanations. After normalizing AI-based explanations (between 0 and 1), we introduced a threshold such that frames with explanations that exceed this threshold are considered important, and unimportant otherwise.

For each threshold, we calculated the precision; the proportion of frames deemed important by the AI system which are also identified as important by the human, and the recall; the proportion of all frames identified as important by the human which the AI system also identified as important. By iterating over a range of thresholds, we can generate a precision-recall curve.

The precision-recall curve reflects the trade-off between the precision of AI-based explanations: the proportion of frames identified as important by the AI system which are actually important, and the recall of such explanations: the proportion of all important frames identified as such by the AI system. For example, recall=1 implies that 100% of the frames identified as important by a human are also identified as such by the AI system. However, an imperfect system can only achieve this outcome by flagging all frames in a video sample as important, irrespective of their ground-truth importance. Naturally, this is an undesirable outcome as the resultant feedback would no longer be temporally-localized, and would thus be less informative. We use the area under the precision-recall curve (AUPRC) as a measure of the reliability of AI-based explanations, as reported in previous studies[25].

**Metrics for evaluating the bias of explanations.** Algorithmic bias is often defined as a discrepancy in the performance of an AI system across sub-cohorts of stakeholders. In this study, we define explanation bias as a discrepancy in the reliability of AI-based explanations across sub-cohorts of surgeons. The intuition is that such a discrepancy implies that a particular sub-cohort of surgeons would systematically receive less reliable feedback, and is thus disadvantaged at a greater rate than other sub-cohorts. To mitigate this explanation bias, we look to improve the reliability of AI-based explanations generated for the disadvantaged sub-cohort of surgeons, referred to as worst-case AUPRC.

*Choice of surgeon groups.* When dealing with live surgical videos, we measured the explanation bias against surgeons operating on prostate glands with different volumes, prostate glands with different cancer severity levels (Gleason Score), and against surgeons with a different caseload (total number of robotic surgeries performed during their lifetime). To decide on these groups, we

consulted with a urologist (A.J.H) about their relevance and opted for groups whose clinical meta information was most complete in our database in effort to increase the number of samples available for analysis. While it may seem out of the ordinary to define a surgeon group based on the prostate volume of a patient being operated on, we note that the distribution of such volumes can also differ across hospitals (e.g., due to patient demographics) (see Supplementary Fig. 1). When dealing with videos from the training environment, we measured the explanation bias against medical students of a different gender.

*Choice of surgeon sub-cohorts.* When dealing with live surgical videos, we converted each of the surgeon groups into categorical sub-cohorts. Specifically, for the prostate volume group, we decided on the two sub-cohorts of prostate volume ≤ 49 ml and > 49 ml. We chose this for practical reasons as it was the population median of the patients at USC, and thereby providing us with a relatively balanced number of video samples from each sub-cohort, and for clinical reasons, with some evidence illustrating that operating on prostate glands of a larger size can increase operating times[29,30]. As for the surgeon caseload group, we decided on the two sub-cohorts of caseload ≤100 and > 100, based on previous studies[31].

**SAIS is an AI system for skill assessment.** We recently developed SAIS to decode the intraoperative activity of surgeons based exclusively on surgical videos[10]. Specifically, we demonstrated state-of-the-art performance in assessing the skill-level of surgical activity, such as needle handling and needle driving, across multiple hospitals. In light of these capabilities, we used SAIS as the core AI system throughout this study.

*Components of SAIS.* We outline the basic components of SAIS here and refer readers to the original study for more details[10]. In short, SAIS takes two data modalities as input: RGB frames and optical flow, which measures motion in the field of view over time, and which is derived from neighbouring RGB frames. Spatial information is extracted from each of these frames through a vision transformer pre-trained in a self-supervised manner on ImageNet. To capture the temporal information across frames, SAIS learns the relationship between subsequent frames through an attention mechanism (see next section). Greater attention, or importance, is placed on frames deemed more important for the ultimate skill assessment. Repeating this process for all data modalities, SAIS arrives at modality-specific video representations. SAIS aggregates these representations to arrive at a single video representation that summarizes the content of the video sample. This video representation is then used to output a probability distribution over the two skill categories (low vs. high skill).

*Generating explanations with SAIS.* To summarize a video sample with a single representation, SAIS adopts an approach often observed with transformer networks used for the purpose of classification; it learns a classification token embedding and treats its corresponding representation after the $N$ transformer encoders as the video representation, $v$ (for one of the modalities, e.g., RGB frames). As the attention mechanism still applies to this video representation, we are able to measure its dependence on all frames in the video sample. The higher the dependence on a particular frame, the more important it is for the assessment of surgeon skill. We hypothesized that the temporal relationship between frames at the final layer of the transformer encoder is most strongly associated with the surgeon skill assessment, and as such, we extracted the attention placed on these frames. This method of explanations is referred to as attention in the Results section.

*Training and evaluating SAIS.* To train and evaluate SAIS, we adopted the same strategy presented in the original study[10]. Specifically, we trained SAIS using 10-fold Monte Carlo cross validation on data exclusively from USC. To ensure that we evaluated SAIS in a rigorous manner, each fold was split into a training, validation, and test set where each set contained samples from surgical videos not present in any of the other sets. Having formulated skill assessment as a binary classification task, we balance the number of video samples from each class (low vs. high-skill) in every data split (training, validation, and test). While doing so during training ensures that the model's performance is not biased towards the majority class, balancing the classes during evaluation (e.g., on the test set) allows for a better appreciation of the evaluation metrics we report. For evaluation on data from other hospitals, we deployed all 10 SAIS models (from the 10 folds). As such, we always report metrics as an average across these 10 folds.

**TWIX is a module for generating AI-based explanations.** Although there exist various ways to incorporate human-based explanations into the learning process of an AI system[24,32], we took inspiration from studies demonstrating the effectiveness of using explanations as a target variable[33–36]. To that end, we propose a strategy entitled training with explanations –TWIX –which explicitly teaches an AI system to generate explanations that match those provided by human experts (see Fig. 2). The intuition is that by incorporating the reasoning used by humans into the learning process of the AI system, it can begin to focus on relevant frames in the video sample and generate explanations that better align with the expectations of humans. To achieve this, we made a simple modification to SAIS (appending a classification module) enabling it to identify the binary importance (important vs. not important) of each frame in the video sample in a supervised manner. Note that TWIX is a general strategy in that it can be used with any architecture, regardless of whether it is attention-based or not.

*Outlining the mechanics of TWIX.* When dealing with the RGB frames of a video sample, SAIS first extracts spatial features from each frame and then captures the temporal relationship between these frames (see previous section for attention mechanism). Upon doing so, SAIS outputs both a single $D$-dimensional video representation, $v \in \mathbb{R}^D$, that summarizes the entire video sample and, importantly, the $D$-dimensional representation of each frame, $h_t \in \mathbb{R}^D$, at time-point $t \in [1, T]$ in a video sample of duration $T$ seconds. For the purpose of surgeon skill assessment, the video representation suffices and is fed into downstream modules. As part of TWIX, however, whose details are presented below, we operate directly on the representation of each frame. Specifically, we keep the core architecture of SAIS unchanged and simply append a classification module, $p_\omega : h_t \in \mathbb{R}^D \rightarrow \hat{y} \in \mathbb{R}$ to map each frame representation, $h_t$, to a scalar frame importance variable, $\hat{y} \in [0, 1]$. The classification module learns which frames are important in a supervised manner based on ground-truth labels, $y$, provided by humans indicating the binary importance of each frame (important vs. not important).

*Training AI systems with TWIX.* We retrain SAIS on data exclusively from USC as reported in the original study[10]. In short, this involves 10-fold Monte Carlo cross-validation where each training fold consists of an equal number of samples from each class (low skill, high skill). The number of samples are provided in Supplementary Table 2. In this study, the difference is that we train SAIS in an end-to-end manner with the TWIX classification module. Specifically, for a single video sample, we optimize the

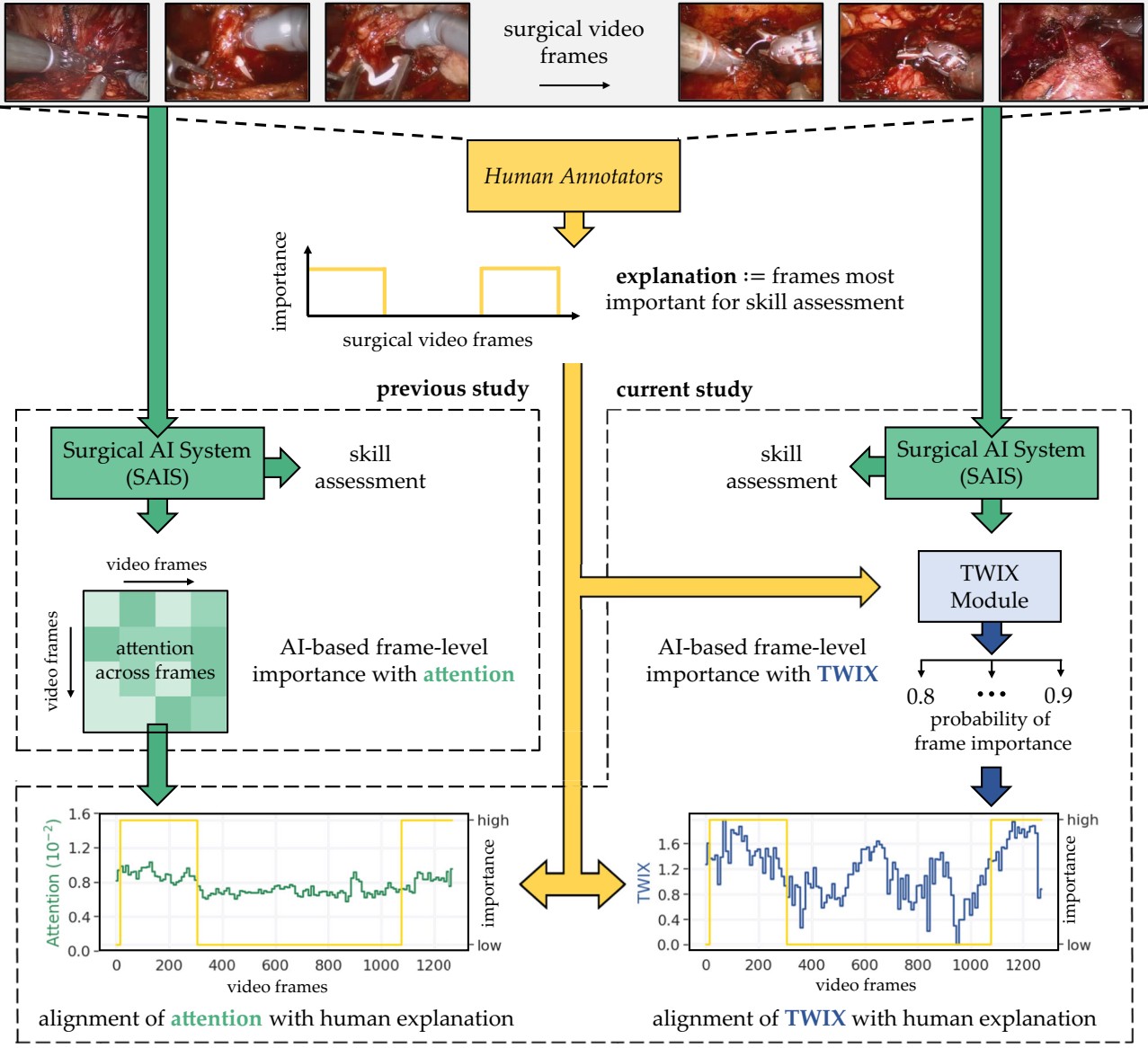

**Fig. 2 Quantifying the alignment of AI-based explanations with human explanations.** A surgical artificial intelligence system (SAIS) can assess the skill of a surgeon based on a surgical video and generate an explanation for such an assessment by highlighting the relative importance of video frames (e.g., via an attention mechanism). Human experts annotate video frames most important for the skill assessment. TWIX is a module which uses human explanations as supervision to explicitly teach an AI system to predict the importance of video frames. We show the alignment of attention and TWIX with human explanations.

supervised InfoNCE loss, $\mathcal{L}_{\text{InfoNCE}}$, reported in the original study, and the binary cross-entropy loss for the classification module, which we refer to as the importance loss, $\mathcal{L}_{\text{importance}}$. Notably, since skill explanation annotations are only provided for the low-skill video samples, the importance loss is only calculated for those video samples.

$$\mathcal{L} = \mathcal{L}_{\text{InfoNCE}}(\theta) + \mathcal{L}_{\text{importance}}(\omega) \qquad (1)$$

$$\mathcal{L}_{\text{importance}}(\omega) = -\sum_{t=1}^{T} (1 - y_t) \log \hat{y}_t + y_t \log(1 - \hat{y}_t)$$

*Predicting frame importance with TWIX.* When SAIS is deployed on unseen data, the classification module, $p_{\omega}$, can now directly output the importance, $\hat{y} \in [0, 1]$, of each frame in a video sample, and thus act as an alternative to the attention mechanism as an indicator of the relative importance of such frames. We refer to this method as TWIX throughout the paper.

Note that the attention mechanism is still a core element of SAIS and continues to capture the temporal relationships between frames irrespective of whether TWIX is adopted or not. In fact, the method entitled attention (w/ TWIX) in the Results section refers to the attention placed on the frames in a video sample, as per usual, however after having adopted TWIX (i.e., after optimizing eq. 1). While it may seem that evaluating explanations again based on these attention values is redundant (i.e., akin to attention before the adoption of TWIX), we show that such attention values can in fact change as a result of optimizing eq. 1. This makes sense since attention is a function of the parameters $\theta$ which, in turn, are now affected by the inclusion of the importance loss, $\mathcal{L}_{importance}$.

**Reporting summary.** Further information on research design is available in the Nature Portfolio Reporting Summary linked to this article.

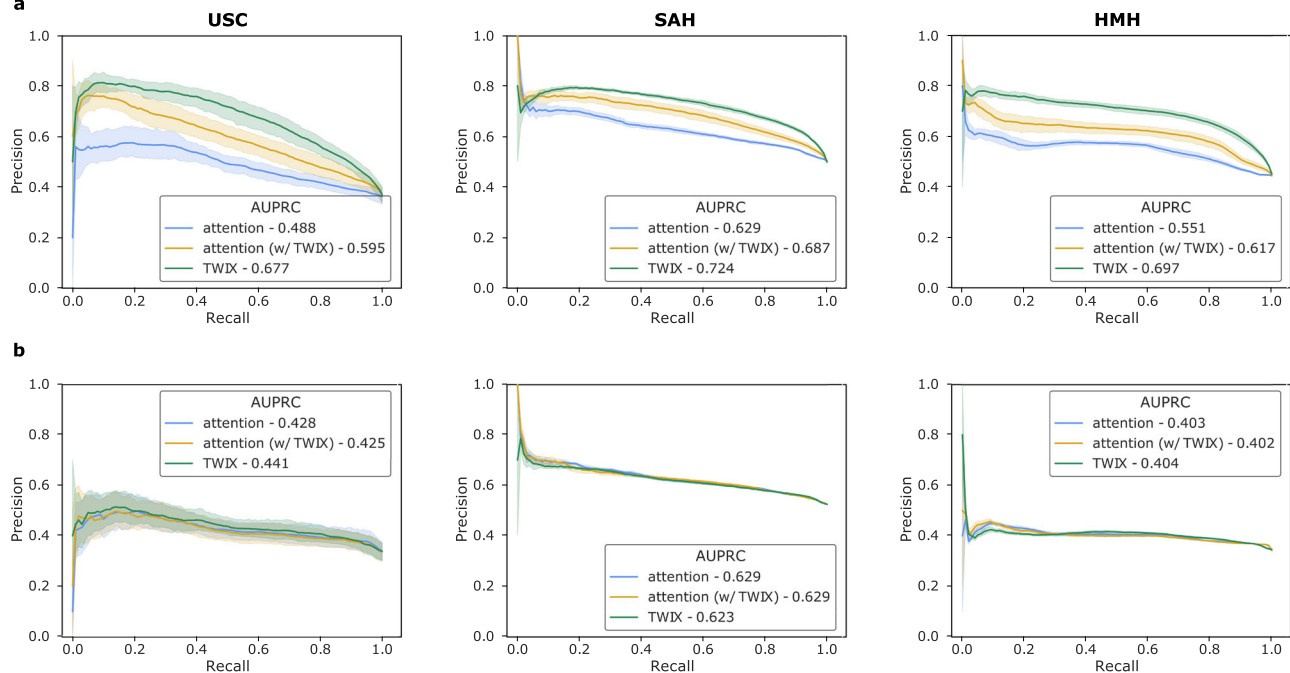

**Fig. 3 TWIX can improve the reliability of AI-based explanations across hospitals.** Precision-recall curves reflecting the alignment of different AI-based explanations with those provided by humans when assessing the skill-level of **a**, needle handling and **b**, needle driving. Note that SAIS is trained exclusively on data from USC and then deployed on data from USC, SAH, and HMH. The solid lines and shaded areas represent the mean and standard deviation, respectively, across 10 Monte Carlo cross-validation folds.

## Results

**SAIS generates explanations that often align with human explanations.** We quantified the reliability of SAIS' explanations (referred to as attention) by comparing them to those generated by human experts. To do so, we first trained and deployed SAIS to perform skill assessment on data from the University of Southern California (USC). We evaluated its explanations for the needle handling and needle driving activities by using the precision-recall (PR) curve[25] (Fig. 3, see Methods for intuition behind PR curves). We found that SAIS' explanations often align, albeit imperfectly, with human explanations. This is evident by AUPRC = 0.488 and 0.428 for the needle handling (Fig. 3a) and needle driving (Fig. 3b) activities, respectively.

**Reliability of explanations is inconsistent across hospitals.** With AI systems often trained on data from one hospital and deployed on data from another hospital, it is important that their behaviour remains consistent across hospitals. Consistent explanations can, for example, improve the trustworthiness of AI systems[37]. To measure this consistency, we trained SAIS on data from USC and deployed it on data from St. Antonius Hospital (SAH), Gronau, Germany and Houston Methodist Hospital (HMH), TX, USA (see Table 1 for number of video samples). We present the precision-recall curves of AI-based explanations for needle handling (Fig. 3a) and needle driving (Fig. 3b). We found that the reliability of SAIS' explanations differ across hospitals. For example, the explanations for needle handling (Fig. 3a) are more reliable when SAIS is deployed on data from SAH and HMH than on data from USC. This is evident by the improved AUPRC = 0.488 → 0.629 at SAH and AUPRC = 0.488 → 0.551 at HMH. We hypothesize that this finding is due to the higher degree of variability in surgical activity depicted in the USC videos relative to that in the other hospitals. This variability might be driven by the larger number of novice surgeons who can exhibit a wider range of surgical activity compared to expert surgeons.

**SAIS exhibits explanation bias against surgeons.** We also investigated whether AI-based explanations are equally reliable for different sub-cohorts of surgeons within the same hospital (Fig. 4, see Supplementary Tables 3-4 for number of samples). We found that SAIS exhibits an explanation bias against surgeon sub-cohorts, whereby its explanations are more reliable for one sub-cohort than another. This is evident, for example, when SAIS assessed the skill-level of needle handling (Fig. 4a) for surgeons operating on prostate glands of different volumes. Whereas AUPRC ≈ 0.54 for prostate volumes ≤ 49 ml, AUPRC ≈ 0.47 for prostate volumes > 49 ml. We observed a similar explanation bias when SAIS assessed the skill-level of needle driving (Fig. 4b).

**Explanation bias is inconsistent across hospitals.** We were further motivated to investigate whether SAIS' explanation bias was consistent across hospitals. To that end, we trained SAIS on data from USC and deployed it on data from SAH and HMH, stratifying the reliability of its explanations across surgeon sub-cohorts (Fig. 3a,b, Supplementary Tables 5-6 outline number of samples). We found that the explanation bias is inconsistent across hospitals. For example, when SAIS assessed the skill-level of needle handling (Fig. 4a) and focusing on surgeons operating on prostate glands of different volumes, we observed an explanation bias at USC (AUPRC ≈ 0.54 vs. 0.47), no bias at SAH (AUPRC ≈ 0.63 vs. 0.63), and an explanation bias against the opposite sub-cohort at HMH (AUPRC ≈ 0.58 vs. 0.63). We found a similarly inconsistent explanation bias for the remaining surgeon groups and even when SAIS assessed the skill-level of needle driving (Fig. 4b).

**TWIX improves reliability of AI-based explanations across hospitals.** Having demonstrated that SAIS' explanations often align, albeit imperfectly, with human explanations, we set out to improve their reliability. We hypothesized that by using human explanations as supervision, we can explicitly teach SAIS to

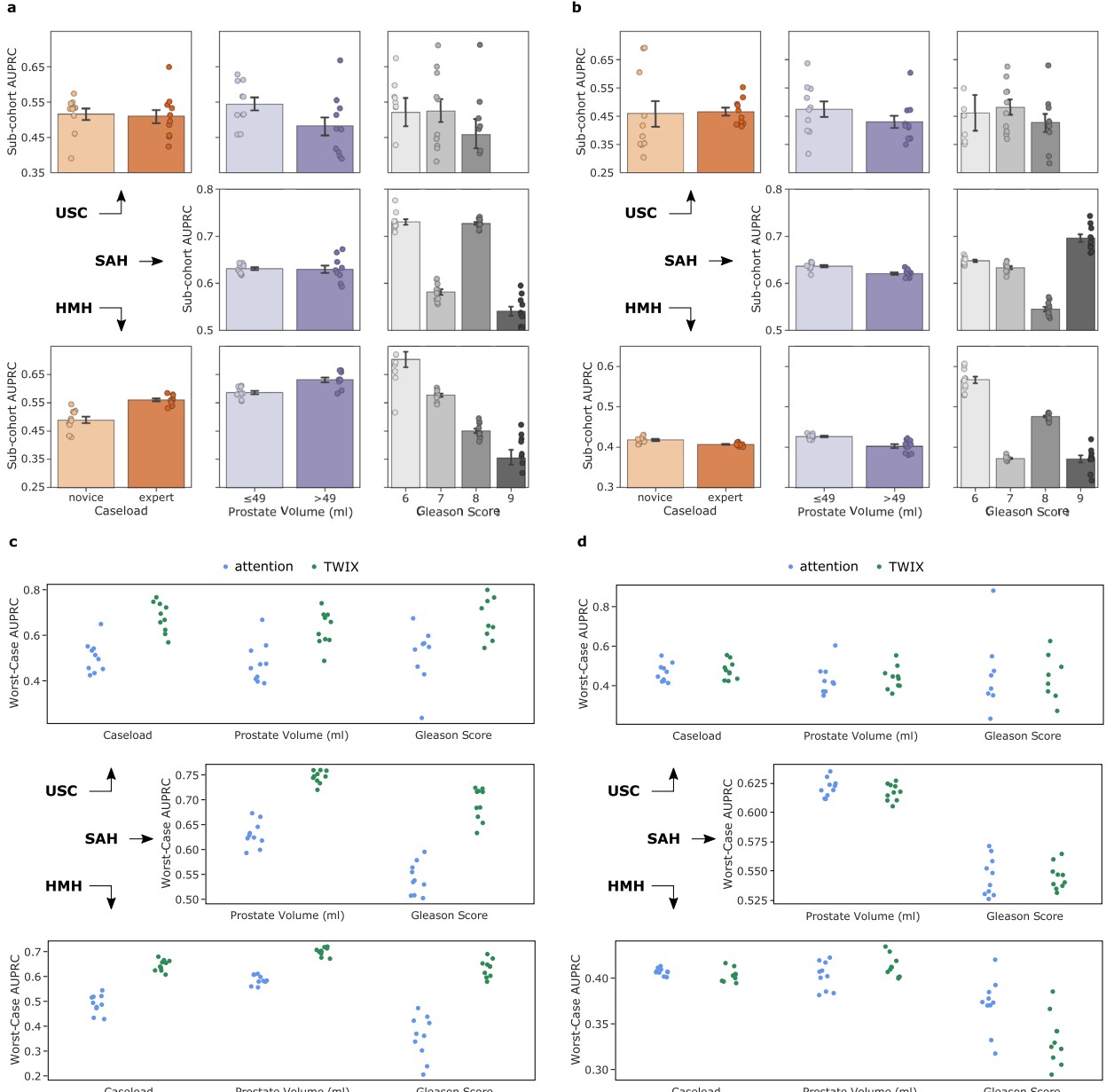

**Fig. 4 TWIX effectively mitigates explanation bias exhibited by SAIS against surgeons.** Reliability of attention-based explanations stratified across surgeon sub-cohorts when assessing the skill-level of **a**, needle handling and **b**, needle driving (see Supplementary Tables 3-6 for number of samples in each sub-cohort). We do not report caseload for SAH due to insufficient samples from one sub-cohort. Effect of TWIX on the reliability of AI-based explanations for the disadvantaged surgeon sub-cohort (worst-case AUPRC) when assessing the skill-level of **c**, needle handling and **d**, needle driving. AI-based explanations come in the form of attention placed on frames by SAIS or through the direct estimate of frame importance by TWIX (see Methods). We do not report caseload for SAH due to insufficient samples from one sub-cohort. Note that SAIS is trained exclusively on data from USC and then deployed on data from USC, SAH, and HMH. Results are an average across 10 Monte Carlo cross-validation folds, and errors bars reflect the 95% confidence interval.

generate explanations that more closely align with human explanations (Fig. 2, right column). This strategy which we refer to as training with explanations –TWIX –directly estimates the importance of each video frame (see Methods). We trained SAIS to assess the skill-level of needle handling and needle driving, while adopting TWIX, on data exclusively from USC and deployed it on data from USC, SAH, and HMH. We present the reliability of AI-based explanations which take on the form of either the attention placed on frames (attention w/ TWIX) or the direct estimate of the importance of frames (TWIX) (Fig. 3).

We found that TWIX improves the reliability of SAIS' attention-based explanations across hospitals. This is evident by the higher AUPRC achieved by attention w/ TWIX. For example, when SAIS assessed the skill-level of needle handling (Fig. 3a), the reliability of attention AUPRC = 0.488 → 0.595 at USC, AUPRC = 0.629 → 0.687 at SAH, and AUPRC = 0.551 → 0.617 at HMH. We did not observe such an improvement when SAIS assessed the skill-level of needle driving (Fig. 3b). One hypothesis for this is that needle driving exhibits a greater degree of variability than needle handling, and therefore assessing its skill level may require the AI system to focus on a more diverse range

of video frames. We also found that TWIX can be more reliable than attention-based explanations, as evident by its relatively higher AUPRC. For example, AUPRC = 0.595 → 0.677 at USC, AUPRC = 0.629 → 0.724 at SAH, and AUPRC = 0.551 → 0.697 at HMH (Fig. 3a). Although TWIX had a minimal benefit on the reliability of explanations when SAIS assessed the skill-level of needle driving (Fig. 3b), it still improved skill assessment performance (see next sections).

**TWIX can effectively mitigate explanation bias across hospitals**. With TWIX improving the reliability of AI-based explanations on average for all surgeons, we wanted to identify if these improvements also applied to the surgeon sub-cohort(s) that previously experienced an explanation bias. Such an improvement would translate to a mitigation in the bias. To do so, we quantified the reliability of AI-based explanations for the disadvantaged sub-cohort of surgeons (worst-case AUPRC) after having adopted TWIX when assessing the skill-level of needle handling (Fig. 4c) and needle driving (Fig. 4d).

We found that TWIX effectively mitigates the explanation bias across hospitals. This is evident by the marked increased in the reliability of SAIS' explanations when assessing the skill-level of needle handling (Fig. 4c) for the previously disadvantaged sub-cohort of surgeons. For example, focusing on surgeons operating on prostate glands of different volumes, the worst-case AUPRC ≈ 0.50 → 0.60 at USC, AUPRC ≈ 0.62 → 0.75 at SAH, and AUPRC ≈ 0.64 → 0.80 at HMH. We observed similarly effective bias mitigation for the remaining surgeon groups. On the other hand, we found that TWIX was not as effective in mitigating the explanation bias when SAIS assessed the skill-level of needle driving (Fig. 4d). We believe that such lack of improvement is due to the high degree of variability in needle driving, implying that the importance of frames in one video sample may not transfer to that in another sample.

**TWIX often improves AI-based skill assessments across hospitals**. Although TWIX was designed to better align AI-based explanations with human explanations, we hypothesized that it might also improve the performance of AI skill assessment systems. The intuition is that by learning to focus on the relevant aspects of the video, SAIS is less likely to latch onto spurious features. To investigate this, we present the performance of SAIS, both with and without TWIX (w/o TWIX), when assessing the skill-level of needle handling and needle driving (Table 2).

As expected, we found that TWIX improves AI-based skill assessments across hospitals. This is evident by the higher AUC values with TWIX than without it. For example, when SAIS assessed the skill-level of needle driving (Table 2), it achieved AUC = 0.822 → 0.850 at USC, AUC = 0.800 → 0.837 at SAH, and AUC = 0.728 → 0.757 at HMH. These findings illustrate that TWIX, which was adopted when SAIS was trained on data exclusively from USC, can still have positive ramifications on performance even when SAIS is deployed on data from an entirely different hospital. In the case of needle handling, we continued to observe the benefits of TWIX on performance, albeit more marginally.

**Ablation study**. Throughout this study, we used the same configuration (e.g., data modalities, problem setup) of SAIS as that presented in the original study[10]. This was motivated by the promising capabilities demonstrated by SAIS and its impending deployment for the provision of feedback. Here, we show how variants of SAIS affect the reliability of explanations (Fig. 5a) and the explanation bias (Fig. 5b), and whether TWIX continues to confer benefits in such settings. Specifically, in addition to

**Table 2 TWIX often improves AI-based skill assessments across hospitals.**

| Skill | Hospital | w/o TWIX | TWIX |
|---|---|---|---|
| needle handling | USC | 0.849 (0.06) | **0.859 (0.05)** |
| | SAH | 0.873 (0.24) | **0.885 (0.02)** |
| | HMH | 0.795 (0.19) | 0.794 (0.03) |
| needle driving | USC | 0.822 (0.05) | **0.850 (0.04)** |
| | SAH | 0.800 (0.04) | **0.837 (0.03)** |
| | HMH | 0.728 (0.05) | **0.757 (0.03)** |

Effect of TWIX on SAIS' ability to assess the skill-level of needle handling and needle driving. Values in bold reflect improvements in performance. Note that SAIS is trained exclusively on data from USC and then deployed on data from USC, SAH, and HMH. Results are an average (standard deviation) across 10 Monte Carlo cross-validation folds.

training SAIS as per normal (RGB + Flow), we also withheld a data modality known as optical flow (RGB), and performed multi-class skill assessment (low vs. intermediate vs. high) (Multi-Skill). We found that TWIX consistently improves the reliability of explanations and mitigates the explanation bias irrespective of the experimental setting in which it is deployed. For example, in the Multi-Skill setting, which is becoming an increasingly preferred way to assess surgeons, the average AUPRC = 0.48 → 0.67 (Fig. 5a) and the worst-case AUPRC = 0.50 → 0.68 (Fig. 5b). These findings demonstrate the versatility of TWIX.

**Providing feedback today in training environment**. Our study builds the foundation for the future implementation of AI-augmented surgical training programs. It is very likely that, in the short run, SAIS will be used to assess the skills of surgical trainees and provide them with feedback on their performance. As with practicing surgeons, it is equally important that such trainees are also not disadvantaged by AI systems. We therefore deployed SAIS on videos from a training environment to assess, and generate explanations for, the skill-level of the needle handling activity performed by participants in control of the same robot otherwise used in live surgical procedures (see Methods) (Fig. 5, c-f).

We discovered that our findings from when SAIS was deployed on video samples from live surgical procedures transfer to the training environment. We found that AI-based explanations often align with those provided by human experts, and that TWIX enhances the reliability of these explanations (Fig. 5c). New to this setting, we found that SAIS exhibits an explanation bias against male surgical trainees (Fig. 5d), an analysis typically precluded by the imbalance in the gender demographics of urology surgeons (national average: > 90% male[38]). Consistent with previous findings, we found that TWIX mitigates the explanation bias, as evident by the improvement in the worst-case AUPRC (Fig. 5e), and improves SAIS' skill assessment capabilities, with an improvement in the AUC (Fig. 5f).

## Discussion
Surgical AI systems can now reliably assess surgeon skills while simultaneously generating an explanation for their assessments. With such explanations likely to inform the provision of feedback to surgeons, it is critical that they align with the expectations of humans and treat all surgeons fairly. However, it has remained an open question whether AI-based explanations exhibit these characteristics.

In this study, we quantified the reliability of explanations generated by surgical AI systems by comparing them to human explanations, and investigated whether such systems generate different quality explanations for different surgeon sub-cohorts.

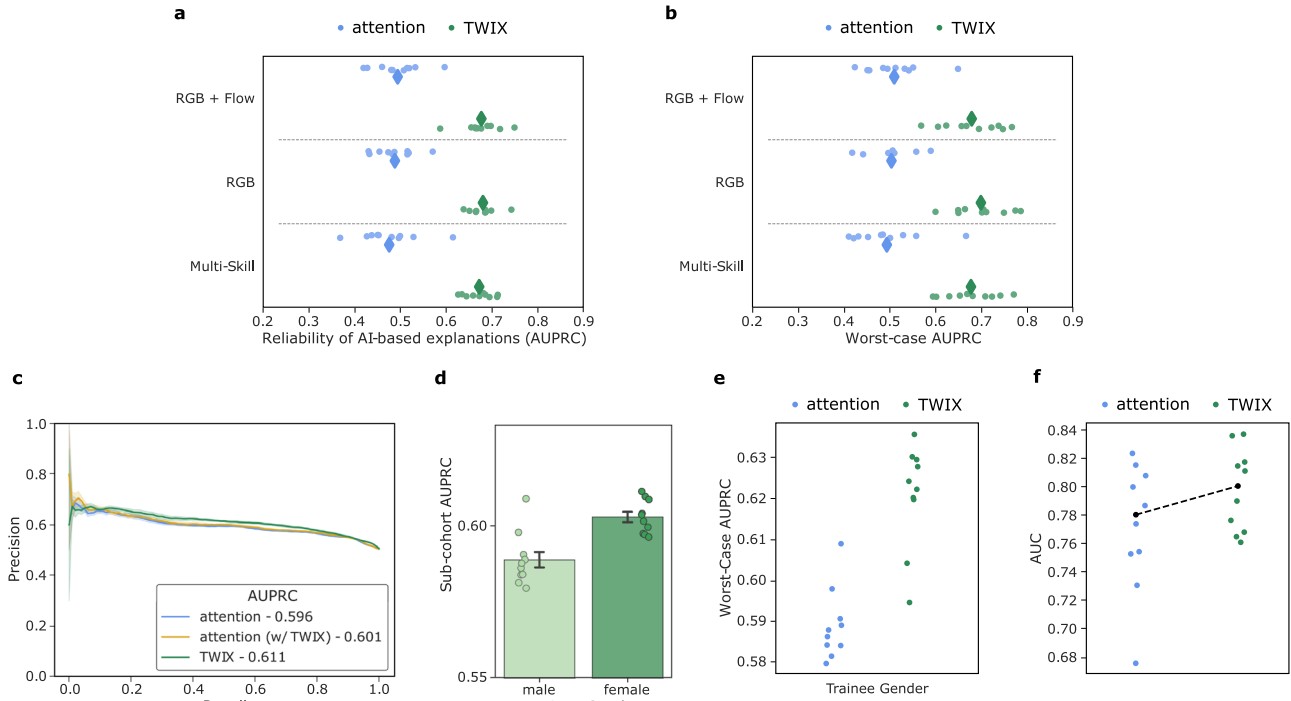

**Fig. 5 TWIX' benefits persist across different experimental settings.** We present the effect of TWIX, in different experimental settings (ablation studies), on **a**, the reliability of explanations generated by SAIS, quantified via the AUPRC, and **b**, the explanation bias, quantified via improvements in the worst-case AUPRC (see Supplementary Tables 3-6 for number of samples in each sub-cohort). The default experimental setting is RGB + Flow and was used throughout this study. Other settings include withholding optical flow from SAIS (RGB) and formulating a multi-class skill assessment task (Multi-Skill). **c**–**f** SAIS can be used today to provide feedback to surgical trainees. **c** AI-based explanations often align with those provided by human experts. **d** SAIS exhibits an explanation bias against male surgical trainees. **e** TWIX mitigates the explanation bias by improving the reliability of explanations provided to male surgical trainees and **f** improves SAIS' performance in assessing the skill-level of needle handling. Note that SAIS is trained exclusively on live data from USC and then deployed on data from the training environment. Results are shown for all 10 Monte Carlo cross-validation folds.

We showed that while AI-based explanations often align with those generated by humans, they can exhibit a bias against surgeon sub-cohorts. To remedy this, we proposed a strategy –TWIX –which uses human explanations as supervision to explicitly teach an AI system to highlight important video frames. We demonstrated that TWIX can improve the reliability and fairness of AI-based explanations, and the overall performance of AI skill assessment systems.

Our study addresses several open questions in the literature. First, the degree of alignment between AI-based explanations and human explanations, and thus their reliability, has thus far remained unknown for video-based surgical AI systems. In previous work, AI-based explanations are often evaluated based on how effectively they guide a human in identifying the content of an image[39,40] and facilitate the detection of errors committed by a model[41,42]. Second, it has also remained unknown whether AI-based explanations exhibit a bias, where their reliability differs across surgeon sub-cohorts. Although preliminary studies have begun to explore the intersection of bias and explanations[26,27,43], they do not leverage human expert explanations, are limited to non-surgical domains, and do not present findings for video-based AI systems. Third, the development of a strategy that consistently improves the reliability and fairness of explanations has been underexplored. Although previous studies have incorporated human explanations into the algorithmic learning process[25,44], they are primarily limited to the discipline of natural language processing and, importantly, do not demonstrate its effectiveness in also improving the fairness of AI-based explanations.

Without first quantifying the reliability and fairness of AI-based explanations, it becomes difficult to evaluate the preparedness of an

AI system for the provision of feedback to surgeons. The implications of misguided feedback can be grave, affecting both surgeons and the patients they eventually operate on. From the surgeon's perspective, receiving unreliable feedback can hinder their professional development, unnecessarily lengthen their learning curve, and prevent them from mastering surgical technical skills. These are acutely problematic given that learning curves for certain procedures can span up to 1000 surgeries[45] and that surgeon performance correlates with postoperative patient outcomes[46,47]. Quantifying the discrepancy in the quality of feedback is equally important. A discrepancy, which we referred to as an explanation bias, results in the provision of feedback that is more reliable for one surgeon sub-cohort than another. Given that feedback can accelerate a surgeon's acquisition of skills, an explanation bias can unintentionally widen the disparity in the skill-set of surgeons. Combined, these implications can complicate the ethical integration of AI systems into surgical training and surgeon credentialing programs. Nonetheless, we believe our framework for quantifying and subsequently improving the alignment of AI-based explanations can benefit other disciplines involving assessments and feedback based on videos, such as childhood education[48] and workplace training[49].

There are certain limitations to our study. We have only measured the reliability of explanations and the effectiveness of TWIX on a single type of surgical activity, namely suturing. However, surgeons must often master a suite of technical skills, including tissue dissection, to proficiently and independently complete an entire surgical procedure. An AI-augmented surgical training program will likely benefit from reliable assessments of distinct surgical activities and corresponding explanations of

those assessments. Furthermore, TWIX requires human-based explanations, which, in the best-case scenario, are difficult and time-consuming to retrieve from experts and, in the worst-case scenario, ambiguous and subjective to provide. Our explanation annotations avoided this latter scenario since they were dependent on a strict set of criteria[28] associated with both visual and motion cues present in the surgical videos. We therefore believe that our approach can be useful in other settings which share these characteristics; where expectations from an AI system can be codified by humans.

We have also made the assumption that AI-based explanations are considered reliable only when they align with human explanations. This interpretation has two potential drawbacks. First, it overlooks the myriad ways in which explanations can be viewed as reliable. For example, they may align with a time period of high blood loss during surgery, which could be consistent with poor postoperative patient outcomes. Evaluating explanations in this way is promising as it would obviate the need for ground-truth human explanations. Instead, the ground-truth importance of video frames can be derived from the context of the surgery (e.g., what and where surgical activity is taking place), which can automatically be decoded by systems like SAIS. Second, constraining AI-based explanations to match human explanations overlooks their promise for the discovery of novel aberrant (or optimal) surgeon behaviour, contributing to the scientific body of knowledge and informing future surgical protocols. Although such discovery is beyond the scope of the present work, it is likely to yield value, for example, when associating intraoperative surgical activity with postoperative patient outcomes.

Several open questions remain unaddressed. First, it remains unknown whether SAIS' explanations accelerate the acquisition of skills by surgical trainees. To investigate this, we plan to conduct a prospective trial amongst medical students in a controlled training environment. Second, despite attempts to define optimal feedback[50,51], in which explanations play an essential role[52], its embodiment remains elusive. In pursuit of that definition, recent frameworks such as the feedback triangle[53] may hold promise, emphasizing the cognitive[54,55], affective, and structural dimensions of feedback. Third, while we have demonstrated that SAIS generates explanations whose reliability differs for different surgeon sub-cohorts, it remains to be seen whether this discrepancy will result in notable harmful consequences. After all, a discrepancy may only translate to an explanation bias if it is unjustified and harmful to surgeons[56].

Surgical training programs continue to adopt the 20th century Halstedian model of "see one, do one, teach one"[57] in reference to learning how to perform surgical procedures. In contrast, AI-augmented surgical training programs can democratize the acquisition of surgical skills on a global scale[58,59] and improve the long-term postoperative outcomes of patients.

## Data availability
As the data contain protected health information, the videos of live surgical procedures and the patients' corresponding demographic information from the University of Southern California, St. Antonius Hospital, and Houston Methodist Hospital are not publicly available. However, since the data from the training environment do not involve patients, those videos and annotations are available on Zenodo (https://zenodo.org/record/7221656#.Y-ZIfi_MI2y) upon reasonable request from the authors. Source data for Fig. 1 is in Supplementary Data 1. Source data for Fig. 3 is in Supplementary Data 2. Source data for Fig. 4 is in Supplementary Data 3 and 4. Source data for Fig. 5 is in Supplementary Data 5.

## Code availability
While SAIS, the underlying AI system, can be accessed at https://github.com/danikiyasseh/SAIS, the code for the existing study can be found at https://github.com/danikiyasseh/TWIX.

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

## Acknowledgements

Research reported in this publication was supported by the National Cancer Institute under Award No. R01CA251579-01A1.

## Author contributions

D.K. contributed to the conception of the study and the study design, developed the deep learning models, and wrote the manuscript. J.L. collected the data from the training environment. D.K., J.L., T.H., and M.O. provided annotations for the video samples. D.A.D. provided feedback on the manuscript. C.W. collected data from St. Antonius-Hospital and B.J.M. collected data from Houston Methodist Hospital, and provided feedback on the manuscript. A.J.H., and A.A. provided supervision and contributed to edits of the manuscript.

## Competing interests

The authors declare the following competing interests: D.K. is a paid consultant of Flatiron Health and an employee of Vicarious Surgical. C.W. is a paid consultant of Intuitive Surgical. A.A. is an employee of Nvidia. A.J.H is a consultant of Intuitive Surgical. The remaining authors declare no competing interests.
