## [Peer Review File · Communications Medicine]

Reviewers' comments:

Reviewer #1 (Remarks to the Author):

This paper is focused on the use of AI models on surgical video with the purpose of replicating human assessment of surgical skill.

The main methodology used in the paper is based on prior work, either from the ML/CV community or from studies from the authors' groups. This is fine as this is a detailed analysis of the application to a multi-centre dataset. There is solid rigour and thinking behind the analysis and the explanations/narrative in the work.

One significant drawback from the work is that there is no plan to release neither the videos nor the annotations. This both makes the contribution exclusive to this particular paper and finding and does not enable further work (which is pretty much guaranteed to be needed) to further propel the field.

The above aside, this is excellent work, and it much needed in this area. Yet there are some things to potentially discuss and specifically to unpick a bit more clearly in the text. For example:

- The data has a huge number of participants. Seems like the average number of procedures (real) per surgeon is about 5 though unlikely to be distributed equally. Isn't this an aspect to investigate? Like what is the effect of explainability across individuals, not across institutions?
- While the above is addressed in part by the bias across surgeons section, it feels a bit like grouping results and not giving confidence on the interpretations.
- Mixing the student and training studies with the real data studies seems confusing for me.
- Why not train on the other datasets too and validate the other way around, e.g. test on the USC data, rather than train on it. Does the same explanation stand up?
- Did you consider ablation studies?
- Technically, it does not seem surprising that injecting more information, improves the explainability. Would this consistently apply if swapping the data labelling methodology, e.g. not clipping videos, etc. How about if not using flow but using 3D networks? How about kinematics?
- I would find it helpful to have a full explanation of all the data. Either pictorial or with graphs. How many samples from each video; the effect on individual videos, etc.

Reviewer #2 (Remarks to the Author):

Many thanks for the invitation to review this paper. The authors should be commended on presenting this paper on an important and highly necessary topic within the field of automated skill assessment.

The authors present impressive and exciting results, however my concerns centre around the dataset used and specifically how ground truth labels were determined which may call into question the reliability of the results presented.

1. Concerning skill assessment annotation, what were the background of the skill annotators (was there any clinical background?). Were any videos marked by experts? What was the justification chosen for 80% agreement as a threshold for adequate training?

2. Why was a binary classification of skill assessment chosen when the original cited assessment tool was on a 3-point Likert scale?

3. How was validation of assessors determination of important periods of video clips determined? Competency in assessment seems to be determined only by capability of assessing high vs low skill.

4. Were videos double assessed? What was the interrater variability of determination of critical periods among the cohort of assessors? Secondly, vs expert surgeons.

5. Could the authors further clarify the distribution of the human raters' perceived critical timestamps? Is there significant weighting to the first x% of the video or was this relatively equally distributed?

6. I question the combination of the use of medical students performing a task in a simulated environment to surgeons performing the steps in live surgery? Given the fact that the authors ultimately chose to focus on low skill participants, this will have constituted a significant proportion of the final dataset. In my opinion, limited focus is given to the justification of this crucial methodological decision.

7. More could be emphasised within the discussion around future clinical and training implications of this technology – should future

aims be to extend beyond highlighting critical video frames to eg. providing narrative feedback

Reviewer #3 (Remarks to the Author):

In this project the authors address the very important task of using AI to assess surgical skill. To this end they collected data from multiple sites, developed a method for ranking performance and developed AI tools to identify skill. I believe that in general this is a very important project that may have a significant impact on the training and assessing of surgical skill.

In this specific study the authors evaluate the reliability of explanations of their AI algorithms, which are important tasks. My first main concern with this manuscript is that I found it hard to follow, and it took me several times until I understood their main contribution. I think the main issue is with the introduction which does not lay the foundation to what is done in the manuscript and what has been presented in other manuscripts. If I understood correction the authors define “explanation” as the ability to show which part of the data is the most important to reach the conclusion. For example if the video is 60 seconds long, highlight the 10 most important seconds. It might be my personal bias, but when I hear the word explanation, I think of more specific explanations such as “you are not holding the needle correctly”. Nevertheless, the authors should be very explicit regarding their definition of explanation in this context.

In essence, the authors compare the ability of general attention model to identify the most important part of the video clip and compare it to a model that is provided with explicit labels regarding the important parts of the video, this was very hard to understand. In addition, the authors mention SAIS and TWIX. However, they do not mention their source in the introduction. Only in the methods section it was made clear that SAIS was developed by the authors and presented in reference [11]. It isn't clear to me where TWIX is described properly. The authors show that TWIX provides better explanation. However, this is not surprising since it receives the explicit labels. I think that from an algorithmic point of view, perhaps the fact that SAIS was able to achieve partial explanation is more impressive, since it is an unsupervised task which achieves nice results. I believe the introduction should be revised. It should be clear what was done in previous studies (by the authors) and what is new in this study. In addition, it should include better definitions.

On the other side the rest of the paper is a bit long and if possible, I would recommend shortening it I think the authors repeat sentences.

I think the paper should be re-written, the introduction should provide a better discretion of where we are heading and the rest should be shorter.

Some smaller comments:

I think they might be able to combine Figure 3 & 4 to one figure.

I find it very surprising that in figure 2, USC has lower results considering the fact the model was trained using USC.

We would like to thank the reviewers for taking the time and effort to review our manuscript and for providing us with valuable feedback. We address your comments below.

We would also like to mention that our previous study, in which we develop SAIS (the AI system underpinning this current study), has since been accepted at *Nature Biomedical Engineering*.

POINT-BY-POINT RESPONSE

Reviewer 1

Summary

This paper is focused on the use of AI models on surgical video with the purpose of replicating human assessment of surgical skill. The main methodology used in the paper is based on prior work, either from the ML/CV community or from studies from the authors' groups. This is fine as this is a detailed analysis of the application to a multi-centre dataset. There is solid rigour and thinking behind the analysis and the explanations/narrative in the work.

R1 – Comment 1

One significant drawback from the work is that there is no plan to release neither the videos nor the annotations. This both makes the contribution exclusive to this particular paper and finding and does not enable further work (which is pretty much guaranteed to be needed) to further propel the field.

Response to R1 – Comment 1

We had outlined in the Data availability statement (from first manuscript submission) that we plan to release both the raw videos and the annotations for the data from the training environment (with medical students). To facilitate the reproducibility of our findings and propel the field forward, we also plan to share the real surgical videos from USC and their corresponding with researchers on a case-by-case basis. The Data availability statement (page 13) has been updated to reflect this.

Data availability

The videos of live surgical procedures from St. Antonius Hospital and Houston Methodist Hospital are not publicly available. While the live videos and corresponding annotations from the University of Southern California will be made available to researchers on a case-by-case basis, those of medical students in the training environment will be made publicly available.

R1 – Comment 2

The above aside, this is excellent work, and it much needed in this area. Yet there are some things to potentially discuss and specifically to unpick a bit more clearly in the text. For example:

The data has a huge number of participants. Seems like the average number of procedures (real) per surgeon is about 5 though unlikely to be distributed equally. Isn't this an aspect to investigate? Like what is the effect of explainability across individuals, not across institutions?

While the above is addressed in part by the bias across surgeons section, it feels a bit like grouping results and not giving confidence on the interpretations.

Response to R1 – Comment 2

Two of the goals of our study were to (1) quantify the reliability of explanations generated by surgical AI systems and (2) measure the potential discrepancy (bias) in the reliability of explanations across surgeon sub-cohorts (e.g., novices vs. experts).

As with almost any AI system, it is always possible to stratify its performance at the level of an individual (e.g., surgeon). Although it is also possible to stratify the reliability of explanations at the surgeon level, we believe there is greater value, at least in the current scope of our study, to quantify the reliability of explanations at a *more aggregated level* (e.g., at the hospital level). This is because it allows us to examine whether our findings generalize across hospitals, which is often viewed as a rigorous approach to evaluating AI systems and methodologies such as TWIX. It signals to readers that TWIX can indeed learn from human supervision and generalize to held-out datasets, and thus increase its likelihood of adoption by future researchers.

As for the second goal, examining bias at the group level is a common choice made by researchers in the field who investigate algorithmic bias. In this context, and from a practical standpoint, we focus on groups of surgeons, as opposed to individual surgeons, due to the relatively larger number of samples in each group, thereby lending greater confidence to our findings.

R1 – Comment 3

Mixing the student and training studies with the real data studies seems confusing for me.

Response to R1 – Comment 3

SAIS was originally developed to assess the skills of surgeons based on videos of real robotic surgeries. In this study, we demonstrated how SAIS and its explanations have the potential to be used for the provision of surgeon feedback. It is very likely that SAIS will be used, in the short run, to assess the skills of surgical trainees and provide them with feedback on their performance. The imminent use of SAIS for such an application motivated our inclusion of the results from the training environment. It is equally important to ensure that surgical trainees, particularly those upstream to practicing surgeons are not disadvantaged by AI skill assessment systems. We have included this motivation in the section **Results → Providing feedback today in training environment (page 6)**.

Providing feedback today in training environment

Our study builds the foundation for the *future* implementation of AI-augmented surgical training programs. It is very likely that, in the short run, SAIS will be used to assess the

skills of surgical trainees and provide them with feedback on their performance. As with practicing surgeons, it is equally important that such trainees are also not disadvantaged by AI systems. We therefore deployed SAIS on videos from a training environment to assess, and generate explanations for, the skill-level of the needle handling activity performed by participants in control of the same robot otherwise used in surgical procedures (see Methods) (Fig. 6).

R1 – Comment 4

Why not train on the other datasets too and validate the other way around, e.g. test on the USC data, rather than train on it. Does the same explanation stand up?

Response to R1 – Comment 4

SAIS was trained exclusively on data from USC and deployed on held-out datasets from USC, St. Antonius Hospital, and Houston Methodist Hospital. This decision was made primarily because of the larger number of samples from USC relative to the other hospitals (see Table 2 for exact number of video samples). By training on data from USC, SAIS was demonstrated to achieve strong generalization performance, an important prerequisite for evaluating the reliability of AI-based explanations. In other words, quantifying the reliability of AI-based explanations is almost moot if the underlying AI system generalizes poorly.

We do, however, appreciate the reviewer’s comment about whether the “same explanations stand up”. We interpret this statement as broadly referring to whether AI-based explanations are stable or robust to changes in the experimental setup (e.g., different training data, different learning protocols, etc.). To that end, we take the reviewer’s suggestion from Comment 5 (next comment) and **conduct two ablation studies** where we (1) withhold the optical flow data modality when training SAIS and (2) train a multi-class skill assessment variant of SAIS, and quantify the reliability of explanations and the explanation bias in these settings (see Results → Ablation study, page 6, paragraph 1, and Figure 5, page 6). In short, we find that TWIX consistently improves the reliability of explanations and mitigates the explanation bias irrespective of the experimental setting in which it is deployed.

Ablation study

Throughout this study, we used the same configuration of SAIS as that presented in the original study⁸. This was motivated by the promising capabilities demonstrated by SAIS and its impending deployment for the provision of feedback. However, we also investigated, in the form of an ablation study, how variants of SAIS affect the reliability of explanations (Fig. 5a) and the explanation bias (Fig. 5b), and whether TWIX continues to confer benefits in such settings. Specifically, in addition to training SAIS as per normal (RGB + Flow), we also withheld a data modality known as optical flow (RGB), and performed multi-class skill assessment (low vs. intermediate vs. high) (Multi-Skill). We found that TWIX consistently improves the reliability of explanations and mitigates the explanation bias irrespective of the experimental setting in which it is deployed. For example, in the Multi-Skill setting, which is becoming an increasingly preferred way to assess surgeons, the average AUPRC = 0.48 → 0.67 (Fig. 5a) and the worst-case AUPRC = 0.50 → 0.68 (Fig. 5b). These findings demonstrate the versatility of TWIX.

Figure 5. Ablation study - the benefits of TWIX persist across different experimental settings. We present the effect of TWIX, in different experimental settings, on (a) the reliability of explanations generated by SAIS, quantified via the AUPRC, and (b) the explanation bias, quantified via improvements in the worst-case AUPRC. The default experimental setting is RGB + Flow and was used throughout this study. Other settings include withholding optical flow from SAIS (RGB) and formulating a multi-class skill assessment task (Multi-Skill). Note that SAIS is trained to perform needle handling skill assessment exclusively on data from USC. Here, it is also deployed on data from USC. Results are shown for all 10 Monte Carlo cross-validation folds.

R1 – Comment 5

Did you consider ablation studies?

Response to R1 – Comment 5

Please see Response to R1 – Comment 4

R1 – Comment 6

Technically, it does not seem surprising that injecting more information, improves the explainability. Would this consistently apply if swapping the data labelling methodology, e.g. not clipping videos, etc. How about if not using flow but using 3D networks? How about kinematics?

Response to R1 – Comment 6

Although supervising the TWIX module with human explanations was expected to improve the reliability of AI-based explanations, this was not guaranteed to occur. Specifically, an AI system that is presented with supervised ground-truth labels must *learn* from such labels such that it is able to generalize to unseen samples. Our contribution is that we demonstrated that TWIX can indeed learn from human explanations and generalize across videos from three geographically-diverse hospitals.

As for quantifying the reliability of explanations under different experimental settings and variants of SAIS, we conducted a set of ablation studies that are described in **Response to R1 – Comment 4** (see **Results → Ablation study, page 6, paragraph 1**, and **Figure 5, page 6**). In short, we find that TWIX consistently improves the reliability of explanations and mitigates the explanation bias irrespective of the experimental setting in which it is deployed.

In our original paper, in which we introduced the SAIS system, we demonstrated that SAIS outperforms the state-of-the-art 3D convolutional networks (Inception3D or I3D) on a multitude of tasks including surgeon skill assessment. **Please note that these results are in the latest version of our original manuscript (not on arXiv) which has since been accepted at *Nature Biomedical Engineering*.** In light

of SAIS' improved performance in assessing surgeon skills relative to I3D, we do not experiment with I3D (for which obtaining frame-level explanations is non-trivial because of the way it processes volumes of frames). As for incorporating additional data modalities (e.g., kinematics), SAIS is a modular architecture that can accept (and ultimately aggregate) any number of input modalities. If kinematics data are available, which we do not currently have access to, then they can seamlessly be incorporated into the learning process.

R1 – Comment 7

I would find it helpful to have a full explanation of all the data. Either pictorial or with graphs. How many samples from each video; the effect on individual videos, etc.

Response to R1 – Comment 7

In the current manuscript, we had outlined the total number of videos and samples from each hospital and for each skill (needle handling and needle driving) (see Table 2, page 9). Supplementary Note 1 also outlines the number of samples in each surgeon sub-cohort, which are used for the experiments in which we stratify the reliability of explanations across sub-cohorts. A more complete description of the data can be found in the latest version of our original manuscript, which has since been accepted at *Nature Biomedical Engineering*.

Reviewer 2

Summary

Many thanks for the invitation to review this paper. The authors should be commended on presenting this paper on an important and highly necessary topic within the field of automated skill assessment. The authors present impressive and exciting results, however my concerns centre around the dataset used and specifically how ground truth labels were determined which may call into question the reliability of the results presented.

R2 – Comment 1

Concerning skill assessment annotation, what were the background of the skill annotators (was there any clinical background?). Were any videos marked by experts? What was the justification chosen for 80% agreement as a threshold for adequate training?

Response to R2 – Comment 1

To clarify, the skill assessment annotations used in this manuscript were obtained from, and are exactly the same as those used in, the original study describing the development and validation of a surgical AI system for decoding the elements of surgery. **That study has since been accepted at *Nature Biomedical Engineering*.**

In the original study, we assembled a team of trained human raters to annotate video samples with skill assessments based on a previously-developed skill assessment taxonomy (also known as an *end-to-end assessment of suturing expertise* or EASE). EASE was formulated through a rigorous Delphi process which involved five expert surgeons that identified a strict set of criteria for assessing multiple skills related to suturing (e.g., needle handling, needle driving, etc.). Our team of raters comprised medical students and surgical residents who either helped devise the original skill assessment taxonomy themselves or had been intimately aware of the details of the taxonomy.

While video samples were not assessed by attending surgeons, we believe the degree of annotation noise is limited for the following reasons. First, EASE outlines a strict set of criteria related to the visual and motion content reflected in a video sample, thereby making it straightforward to identify whether such criteria are satisfied (or violated) upon watching a video sample. This reduces the level of expertise that a rater must ordinarily have in order to annotate a video sample. Second, the raters involved in the annotation process were either a part of the development of the EASE taxonomy or intimately aware of its details. This implied that they were comfortable with the criteria outlined in EASE. Third, and understanding that raters can be imperfect, we subjected them to a training process whereby raters were provided with a training set of video samples and asked to annotate them independently of one another. This process continued until the agreement of their annotations, which was quantified via inter-rater reliability, exceeded 80%. We chose this threshold based on (a) the level of agreement first reported in the study developing the EASE taxonomy and (b) an appreciation that natural variability is likely to exist from one rater to the next in, for example, the amount of attention they place on certain content within a video sample (**Methods → Surgical video samples and annotations → Skill assessment annotations (page 9 – 10)**)

Skill assessment annotations

We assembled a team of trained human raters to annotate video samples with skill assessments based on a previously-developed skill assessment taxonomy (also known as an end-to-end assessment of suturing expertise or EASE⁵¹). EASE was formulated through a rigorous Delphi process which involved five expert surgeons that identified a strict set of criteria for assessing multiple skills related to suturing (e.g., needle handling, needle driving, etc.). Our team of raters comprised medical students and surgical residents who either helped devise the original skill assessment taxonomy themselves or had been intimately aware of its details.

Mitigating noise in skill assessment annotations While video samples were not assessed by attending surgeons, we believe the degree of annotation noise is limited for the following reasons. First, EASE outlines a strict set of criteria related to the visual and motion content reflected in a video sample, thereby making it straightforward to identify whether such criteria are satisfied (or violated) upon watching a video sample. This reduces the level of expertise that a rater must ordinarily have in order to annotate a video sample. Second, the raters involved in the annotation process were either a part of the development of the EASE taxonomy or intimately aware of its details. This implied that they were comfortable with the criteria outlined in EASE. Third, and understanding that raters can be imperfect, we subjected them to a training process whereby raters were provided with a training set of video samples and asked to annotate them independently of one another. This process continued until the agreement of their annotations, which was quantified via inter-rater reliability, exceeded 80%. We chose this threshold based on (a) the level of agreement first reported in the study developing the EASE taxonomy and (b) an appreciation that natural variability is likely to exist from one rater to the next in, for example, the amount of attention they place on certain content within a video sample.

Aggregating skill assessment annotations After completing their training process, raters were asked to annotate the video samples in this study with binary skill assessments (low vs. high skill). In the event of disagreements in the annotations, we followed the same strategy adopted in the original study⁸ where the lowest of all scores is considered as the final annotation.

R2 – Comment 2

Why was a binary classification of skill assessment chosen when the original cited assessment tool was on a 3-point Likert scale?

Response to R2 – Comment 2

While EASE (the skill assessment taxonomy) does outline a set of criteria for classifying skill into three distinct categories (low vs. intermediate vs. high), the family of surgical AI systems (SAIS) which we leverage throughout this study was developed to perform binary skill assessment (low vs. high skill).

That decision was originally made for practical reasons, where we had an insufficient number of video samples annotated as intermediate skill to warrant their inclusion in the learning process of the AI system. We therefore opted to leverage the video samples annotated as low or high skill to develop a binary skill assessment system.

In this study, our use of a binary skill assessment system fits well with our goal of providing feedback for video samples that were annotated as depicting *low skill activity*. The motivation behind our focus on low skill activity is twofold. First, from a practical standpoint, it is relatively more straightforward to provide an explanation annotation for a video sample depicting low skill activity than it is for one depicting high skill activity. This is because human raters simply have to look for segments in the video sample during which one (or more) of the criteria outlined in EASE are violated. Second, from an educational standpoint, studies in the domain of educational psychology have demonstrated that corrective feedback following an error is instrumental to learning [1]. As such, our focus on a low skill activity (akin to an error) provides a ripe opportunity for the provision of feedback. We do appreciate, however, that feedback can also be useful when provided for video samples depicting *high skill activity* (e.g., through positive reinforcement). We leave this as an extension of our work for the future (Methods → Motivation behind focusing on low-skill activity, page 11, paragraph 1).

Motivation behind focusing on low-skill activity

In this study, our goal was to provide feedback for video samples depicting low skill activity. A binary skill assessment system is therefore well aligned with this goal. We focused on low-skill activity for two reasons. First, from a practical standpoint, it is relatively more straightforward to provide an explanation annotation for a video sample depicting low skill activity than it is for one depicting high skill activity. This is because human raters simply have to look for segments in the video sample during which one (or more) of the criteria outlined in EASE are violated. Second, from an educational standpoint, studies in the domain of educational psychology have demonstrated that corrective feedback following an error is instrumental to learning [1]. As such, our focus on a low skill activity (akin to an error) provides a ripe opportunity for the provision of feedback. We do appreciate, however, that feedback can also be useful when provided for video samples depicting high skill activity (e.g., through positive reinforcement). We leave this as an extension of our work for the future. The average duration of the explanation annotations can be found in Table S2.

Having motivated our use of a binary skill assessment system, we also train SAIS to perform multi-class skill assessment (low vs. intermediate vs. high) for the skill of needle handling. In Results → Ablation study (page 6, paragraph 1) and Figure 5 (page 6), we present the reliability of SAIS' explanations in this setting and its explanation bias, before and after using TWIX. In short, we demonstrate that TWIX continues to improve the reliability of explanations and mitigate the explanation bias irrespective of the experimental setting in which it is deployed.

Ablation study

Throughout this study, we used the same configuration of SAIS as that presented in the original study⁸. This was motivated by the promising capabilities demonstrated by SAIS and its impending deployment for the provision of feedback. However, we also investigated, in the form of an ablation study, how variants of SAIS affect the reliability of explanations (Fig. 5a) and the explanation bias (Fig. 5b), and whether TWIX continues to confer benefits in such settings. Specifically, in addition to training SAIS as per normal (RGB + Flow), we also withheld a data modality known as optical flow (RGB), and performed multi-class skill assessment (low vs. intermediate vs. high) (Multi-Skill). We found that TWIX consistently improves the reliability of explanations and mitigates the explanation bias irrespective of the experimental setting in which it is deployed. For example, in the Multi-Skill setting, which is becoming an increasingly preferred way to assess surgeons, the average AUPRC = 0.48 → 0.67 (Fig. 5a) and the worst-case AUPRC = 0.50 → 0.68 (Fig. 5b). These findings demonstrate the versatility of TWIX.

Figure 5. Ablation study - the benefits of TWIX persist across different experimental settings. We present the effect of TWIX, in different experimental settings, on (a) the reliability of explanations generated by SAIS, quantified via the AUPRC, and (b) the explanation bias, quantified via improvements in the worst-case AUPRC. The default experimental setting is RGB + Flow and was used throughout this study. Other settings include withholding optical flow from SAIS (RGB) and formulating a multi-class skill assessment task (Multi-Skill). Note that SAIS is trained to perform needle handling skill assessment exclusively on data from USC. Here, it is also deployed on data from USC. Results are shown for all 10 Monte Carlo cross-validation folds.

R2 – Comment 3

How was validation of assessors determination of important periods of video clips determined? Competency in assessment seems to be determined only by capability of assessing high vs low skill.

Response to R2 – Comment 3

We assembled a team of two trained human raters to annotate each video sample with segments of time (or equivalently, spans of frames) deemed relevant for a particular skill assessment. We define segments of time as relevant if they reflect the strict set of criteria (or their violation) outlined in the skill assessment taxonomy. In practice, we asked raters to exclusively annotate video samples previously tagged as low skill from a previous study. Our motivation for doing so is outlined in the Methods section. For the activity of needle handling, a low skill assessment is characterized by three

or more grasps of the needle by the surgical instrument. For the activity of needle driving, a low skill assessment is characterized by either four or more adjustments of the needle when being driven through tissue or its complete removal from tissue in the opposite direction to which it was inserted. As such, raters had to identify both visual and motion cues in the surgical field of view in order to annotate segments of time as relevant (Methods → Surgical video samples and annotations → Skill explanation annotations, page 9 – 10).

Skill explanation annotations

We assembled a team of two trained human raters to annotate each video sample with segments of time (or equivalently, spans of frames) deemed relevant for a particular skill assessment. We define segments of time as relevant if they reflect the strict set of criteria (or their violation) outlined in the skill assessment taxonomy⁵¹. In practice, we asked raters to exclusively annotate video samples previously tagged as low skill from a previous study (for motivation, see later section). For the activity of needle handling, a low skill assessment is characterized by three or more grasps of the needle by the surgical instrument. For the activity of needle driving, a low skill assessment is characterized by either four or more adjustments of the needle when being driven through tissue or its complete removal from tissue in the opposite direction to which it was inserted. As such, raters had to identify both visual and motion cues in the surgical field of view in order to annotate segments of time as relevant. We reflect important

Before providing such *explanation annotations*, however, the raters underwent a training process akin to the one conducted for skill assessment annotations. First, raters were familiarized with the criteria outlined in the skill assessment taxonomy. In practice, and to mitigate potential noise in the explanation annotations, our assembled team of raters had, in the past, already been involved in providing skill assessment annotations while using the same exact taxonomy. The raters were then provided with a training set of low-skill video samples and asked to independently annotate them with segments of time that they believed were important to that skill assessment. During this time, raters were encouraged to abide by the strict set of criteria outlined in the skill assessment taxonomy. This training process continued until the agreement in their annotations, which was quantified via the intersection over union, exceeded 0.80. This implies that, on average, each segment of time highlighted by one rater exhibited an 80% overlap with that provided by another rater. This value was chosen, as with the skill assessment annotation process, having appreciated that natural variability in the annotation process is likely to occur. Raters may disagree, for example, on when an important segment of time ends even when both of their explanation annotations capture the bulk of the relevant activity (Methods → Surgical video samples and annotations → Skill explanation annotations → Training the raters, page 10, paragraph 4).

Training the raters Before providing such explanation annotations, however, the raters underwent a training process akin to the one conducted for skill assessment annotations. First, raters were familiarized with the criteria outlined in the skill assessment taxonomy. In practice, and to mitigate potential noise in the explanation annotations, our assembled team of raters had, in the past, already been involved in providing skill assessment annotations while using the same exact taxonomy. The raters were then provided with a training set of low-skill video samples and asked to independently annotate them with segments of time that they believed were important to that skill assessment. During this time, raters were encouraged to abide by the strict set of criteria outlined in the skill assessment taxonomy. This training process continued until the agreement in their annotations, which was quantified via the intersection over union, exceeded 0.80. This implies that, on average, each segment of time highlighted by one rater exhibited an 80% overlap with that provided by another rater. This value was chosen, as with the skill assessment annotation process, with the appreciation that natural variability in the annotation process is likely to occur. Raters may disagree, for example, on when an important segment of time ends even when both of their explanation annotations capture the bulk of the relevant activity.

Upon completing the training process, raters were asked to provide explanation annotations for the video samples used in this study. They were informed that each video sample had been annotated in the past as low skill, and were therefore aware of the specific criteria in the taxonomy to look out for. In the event of disagreements in the explanation annotations, we considered the intersection of the annotations. This ensures that we avoid identifying potentially superfluous video frames as relevant and makes us more confident in the segments of time that overlapped amongst the raters' annotations. Although we experimented with other strategies for aggregating the explanation annotations, such as considering their union, we found this to have a minimal effect on our findings (Methods → Surgical video samples and annotations → Skill explanation annotations → Aggregating explanation annotations, page 10, paragraph 4).

Aggregating explanation annotations Upon completing the training process, raters were asked to provide explanation annotations for the video samples used in this study. They were informed that each video sample had been annotated in the past as low skill, and were therefore aware of the specific criteria in the taxonomy to look out for. In the event of disagreements in the explanation annotations, we considered the intersection of the annotations. This ensures that we avoid identifying potentially superfluous video frames as relevant and makes us more confident in the segments of time that overlapped amongst the raters' annotations. Although we experimented with other strategies for aggregating the explanation annotations, such as considering their union, we found this to have a minimal effect on our findings.

R2 – Comment 4

Were videos double assessed? What was the interrater variability of determination of critical periods among the cohort of assessors? Secondly, vs expert surgeons.

Response to R2 – Comment 4

Yes, please see our Response to R2 – Comment 3.

R2 – Comment 5

Could the authors further clarify the distribution of the human raters' perceived critical timestamps? i.e., was there significant weighting to the first x% of the video or was this relatively equally distributed?

Response to R2 – Comment 5

To give readers a better appreciation of the ground-truth explanation annotations, we incorporate the reviewer's suggestion into our manuscript by presenting a heatmap of the explanations over time at the distinct hospitals and for the two skills (needle handling and needle driving). These heatmaps are shown in Figure 7 (Methods → Surgical video samples and annotations → Skill explanation annotations → generating and visualising explanation heatmaps, page 10, paragraph 1).

To generate these heatmaps, we considered unique video samples in the test set of each Monte Carlo fold (10 folds in total). Since each video sample may vary in duration, and to facilitate a comparison of the heatmaps across hospitals, we first normalized the time index of each explanation annotation such that it ranged from 0 (beginning of video sample) to 1 (end of video sample). In the context of needle handling, for example, this translates to the beginning and end of needle handling, respectively. As another example, a value of 0.20 refers to the first 20% of the video sample. We then averaged the explanation annotations, whose values are either 0 (irrelevant frame) or 1 (relevant frame), across the video samples for this normalized time index. We repeated the process for all hospitals and skills (needle handling and needle driving).

Generating and visualising explanation heatmaps Given a video sample 30 seconds in duration, a human rater might annotate the first five seconds (0 – 5 seconds) as most important for the skill assessment. In Table S2, we show that, on average, around 30% of a single video sample is identified as important.

When we normalize the duration of all video samples, with 0 and 1 reflecting the beginning and end of a video sample respectively, we can average all ground-truth explanation annotations for the skill of needle handling (Fig. 7a) and needle driving (Fig. 7b) in each hospital. We find that at USC, for example, the more important time segments for low-skill needle handling tend to occur earlier in the video sample (e.g., in the first 20%).

To generate these explanation heatmaps, we considered unique video samples in the test set of each Monte Carlo fold (10 folds in total). Since each video sample may vary in duration, and to facilitate a comparison of the heatmaps across hospitals, we first normalized the time index of each explanation annotation such that it ranged from 0 (beginning of video sample) to 1 (end of video sample). In the context of needle handling, for example, this translates to the beginning and end of needle handling, respectively. As another example, a value of 0.20 refers to the first 20% of the video sample. We then averaged the explanation annotations, whose values are either 0 (irrelevant frame) or 1 (relevant frame), across the video samples for this normalized time index. We repeated the process for all hospitals and skills (needle handling and needle driving).

Figure 7. Heatmap of the ground-truth explanation annotations across hospitals. We average the explanation annotations for the (a) needle handling and (b) needle driving video samples in the test set of the Monte Carlo folds, and present them over a normalized time index, where 0 and 1 reflect the beginning and end of a video sample, respectively. A darker shade (which ranges from 0 to 1 as per the colour bars) implies that a segment of time is of greater importance.

R2 – Comment 6

I question the combination of the use of medical students performing a task in a simulated environment to surgeons performing the steps in live surgery? Given the fact that the authors ultimately chose to focus on low skill participants, this will have constituted a significant proportion of the final dataset. In my opinion, limited focus is given to the justification of this crucial methodological decision.

Response to R2 – Comment 6

To clarify, we made the decision to focus on explanations associated with low-skill *activity*, and not necessarily low-skill *participants*. The distinction here is that experienced surgeons can still exhibit low-skill activity, according to our previously-developed skill assessment taxonomy. Conversely, medical students and surgical trainees can exhibit high-skill activity. Therefore, we believe that all individuals, irrespective of their experience, can benefit from surgical training and feedback.

We had provided a motivation for exclusively focusing on low-skill activity in the section **Methods → Motivation behind focusing on low skill activity (page 11)**. As we mentioned in **Response to R2 – Comment 2**, we do appreciate that feedback can also be useful when provided for video samples depicting *high skill activity* (e.g., through positive reinforcement). We leave this as an extension of our work for the future.

Motivation behind focusing on low-skill activity

In this study, our goal was to provide feedback for video samples depicting low skill activity. A binary skill assessment system is therefore well aligned with this goal. We focused on low-skill activity for two reasons. First, from a practical standpoint, it is relatively more straightforward to provide an explanation annotation for a video sample depicting low skill activity than it is for one depicting high skill activity. This is because human raters simply have to look for segments in the video sample during which one (or more) of the criteria outlined in EASE are violated. Second, from an educational standpoint, studies in the domain of educational psychology have demonstrated that corrective feedback following an error is instrumental to learning [1]. As such, our focus on a low skill activity (akin to an error) provides a ripe opportunity for the provision of feedback. We do appreciate, however, that feedback can also be useful when provided for video samples depicting high skill activity (e.g., through positive reinforcement). We leave this as an extension of our work for the future. The average duration of the explanation annotations can be found in Table S2.

R2 – Comment 7

More could be emphasised within the discussion around future clinical and training implications of this technology – should future aims be to extend beyond highlighting critical video frames to e.g., providing narrative feedback

Response to R2 – Comment 7

We have now expanded our **Discussion section (page 7, paragraph 3)** to outline, in more depth, the clinical and training implications of our framework and being dependent on AI-based explanations.

Without first quantifying the reliability and fairness of AI-based explanations, it becomes difficult to evaluate the preparedness of an AI system for the provision of surgeon feedback. The implications of misguided feedback can be grave,

affecting both surgeons and the patients they eventually operate on. From the surgeon's perspective, receiving unreliable feedback can hinder their professional development, unnecessarily lengthen their learning curve, and prevent them from mastering surgical technical skills. These are acutely problematic given that learning curves for certain procedures can span up to 1,000 surgeries³⁵ and that surgeon performance correlates with postoperative patient outcomes^{36,37}. Quantifying the discrepancy in the quality of surgeon feedback is equally important. A discrepancy, which we referred to as an explanation bias, implies the provision of feedback that is more reliable for one surgeon sub-cohort than another. Given that feedback can accelerate a surgeon's acquisition of skills, an explanation bias can unintentionally widen the disparity in the skill-set of surgeons. Combined, these implications can complicate the ethical integration of AI systems into surgical training programs. Nonetheless, we believe our framework for quantifying and subsequently improving the alignment of AI-based explanations can benefit other disciplines involving assessments and feedback based on videos, such as childhood education³⁸ and workplace training³⁹.

Reviewer 3

Summary

In this project the authors address the very important task of using AI to assess surgical skill. To this end they collected data from multiple sites, developed a method for ranking performance and developed AI tools to identify skill. I believe that in general this is a very important project that may have a significant impact on the training and assessing of surgical skill.

R3 – Comment 1

In this specific study the authors evaluate the reliability of explanations of their AI algorithms, which are important tasks. My first main concern with this manuscript is that I found it hard to follow, and it took me several times until I understood their main contribution. I think the main issue is with the introduction which does not lay the foundation to what is done in the manuscript and what has been presented in other manuscripts.

Response to R3 – Comment 1

To improve the clarity of the **Introduction (page 1)**, we make the following changes:

- **Paragraph 1** – we clearly introduce SAIS (our previously-developed AI system). This should make it clear that SAIS has already been developed and, in this study, we are experimenting with and building upon it.
- **Paragraph 1** – we include our definition of “explanations” (which the reviewer had correctly understood as highlighting the most important frames in a video)

Introduction

Surgeons seldom receive feedback on how well they perform surgery^{1,2}, despite evidence that it accelerates their acquisition of skills (e.g., suturing)³⁻⁷. Such feedback can be provided by a recently-developed surgical artificial intelligence system (SAIS)⁸ that reliably assesses the skill of a surgeon based on a video of intraoperative activity while simultaneously highlighting frames with pertinent activity^{9,10}. To safely automate the provision of feedback, these highlights, which we refer to as AI-based explanations, should ideally match the expectations of expert surgeons (reliable)^{11,12} and be equally reliable for all surgeons (fair)¹³. *However, it remains an open question whether AI-based explanations are reliable and fair.* If left unchecked, misguided feedback can hinder the professional development of surgeons and unethically disadvantage one surgeon sub-cohort over another (e.g., novices vs. surgeons).

- **Paragraph 2** – we provide a more accessible description of attention scores to allow readers to better understand how SAIS generates explanations.

In disciplines where the attention-based transformer architecture¹⁴ has gained traction, such as natural language processing¹⁵ and protein modelling^{16,17}, attention scores are often treated as a form of explanation. An element with a higher attention score is assumed to be more important than that with a lower score. Similarly, as a transformer-based architecture which operates on videos, SAIS⁸ generates an attention score

- We have modified **Figure 1 (page 2)** to more clearly delineate between what we have done in previous studies and what we focus on in the current study. We believe this modified figure should also help address any potential confusion about the contributions of our study.

R3 – Comment 2

I think the main issue is with the introduction which does not lay the foundation to what is done in the manuscript and what has been presented in other manuscripts.

Response to R3 – Comment 2

Please see our Response to R3 – Comment 1. In short, the modified **Figure 1 (page 2)** delineates what we have done in previous studies from what we focus on in this current study.

R3 – Comment 3

If I understood correction the authors define “explanation” as the ability to show which part of the data is the most important to reach the conclusion. For example if the video is 60 seconds long, highlight the 10 most important seconds. It might be my personal bias, but when I hear the word explanation, I think of more specific explanations such as “you are not holding the needle correctly”. Nevertheless, the authors should be very explicit regarding their definition of explanation in this context. In essence, the authors compare the ability of general attention model to identify the most important part of the video clip and compare it to a model that is provided with explicit labels regarding the important parts of the video, this was very hard to understand.

Response to R3 – Comment 3

We have now explicitly included our definition of “explanations” in the **Introduction section (page 1, paragraph 1)**. By including this definition early in the manuscript, we hope it alleviates any potential confusion about what “explanations” are referring to.

We also visualize these ground-truth explanations, in the form of explanation heatmaps (**Methods → Surgical video samples and annotations → Skill explanation annotations → Generating and visualizing explanation heatmaps, Figure 7, page 10**). This should concretely demonstrate that explanations, in the context of this study, are purely visual and correspond to frames identified as important in a video.

Generating and visualising explanation heatmaps Given a video sample 30 seconds in duration, a human rater might annotate the first five seconds (0 – 5 seconds) as most important for the skill assessment. In Table S2, we show that, on average, around 30% of a single video sample is identified as important.

When we normalize the duration of all video samples, with 0 and 1 reflecting the beginning and end of a video sample respectively, we can average all ground-truth explanation annotations for the skill of needle handling (Fig. 7a) and needle driving (Fig. 7b) in each hospital. We find that at USC, for example, the more important time segments for low-skill needle handling tend to occur earlier in the video sample (e.g., in the first 20%).

To generate these explanation heatmaps, we considered unique video samples in the test set of each Monte Carlo fold (10 folds in total). Since each video sample may vary in duration, and to facilitate a comparison of the heatmaps across hospitals, we first normalized the time index of each explanation annotation such that it ranged from 0 (beginning of video sample) to 1 (end of video sample). In the context of needle handling, for example, this translates to the beginning and end of needle handling, respectively. As another example, a value of 0.20 refers to the first 20% of the video sample. We then averaged the explanation annotations, whose values are either 0 (irrelevant frame) or 1 (relevant frame), across the video samples for this normalized time index. We repeated the process for all hospitals and skills (needle handling and needle driving).

Figure 7. Heatmap of the ground-truth explanation annotations across hospitals. We average the explanation annotations for the (a) needle handling and (b) needle driving video samples in the test set of the Monte Carlo folds, and present them over a normalized time index, where 0 and 1 reflect the beginning and end of a video sample, respectively. A darker shade (which ranges from 0 to 1 as per the colour bars) implies that a segment of time is of greater importance.

R3 – Comment 4

In addition, the authors mention SAIS and TWIX. However, they do not mention their source in the introduction. Only in the methods section it was made clear that SAIS was developed by the authors and presented in reference [11]. It isn't clear to me where TWIX is described properly.

Response to R3 – Comment 4

We now explicitly mention SAIS in the **Introduction (page 1, paragraph 1)** and outline that it was part of a previous study in **Figure 1 (page 2)**. We would also like to mention that our previous study, in which we develop SAIS (the AI system underpinning this current study), has since been accepted at *Nature Biomedical Engineering*.

As for TWIX, we now include a brief description of it in the **Results → TWIX improves reliability of AI-based explanations across hospitals (page 4, paragraph 2)**. A more in-depth description can be found in **Methods → TWIX is a module for generating AI-based explanations (page 12 – 13)**.

TWIX improves reliability of AI-based explanations across hospitals

We hypothesized that by using human explanations as supervision, we can explicitly teach an AI system to generate explanations that more closely align with human explanations (Fig. 1, right column), a strategy we refer to as training with explanations –TWIX (see Methods). Specifically, we trained SAIS, while adopting TWIX, on data exclusively from USC and deployed it on data from USC, SAH, and HMH to assess the skill-level of needle handling and needle driving (Fig. 2). We present AI-based explanations in the form of either the attention placed on frames (attention w/ TWIX) or the direct

R3 – Comment 5

The authors show that TWIX provides better explanation. However, this is not surprising since it receives the explicit labels. I think that from an algorithmic point of view, perhaps the fact that SAIS was able to achieve partial explanation is more impressive, since it is an unsupervised task which achieves nice results.

Response to R3 – Comment 5

Although supervising the TWIX module with human explanations was expected to improve the reliability of AI-based explanations, this was not guaranteed to occur. Specifically, an AI system that is presented with supervised ground-truth labels must *learn* from such labels such that it is able to generalize to unseen samples. Our contribution is that we demonstrated that TWIX can indeed learn from human explanations and generalize across videos from three geographically-diverse hospitals.

As the reviewer points out, SAIS' attention-based explanation (without TWIX) often aligned, albeit imperfectly, with human explanations (as we had outlined in **Results → SAIS generates explanations that often align with human-based explanations, page 3**). This was our first finding and allowed us to quantify the reliability of SAIS' explanations. Up until now, the alignment of AI-based explanations with human explanations in the context of surgical videos had not been investigated. Throughout the rest of the manuscript, we demonstrate, amongst other things, how TWIX can improve upon the reliability of these explanations.

R3 – Comment 6

I believe the introduction should be revised. It should be clear what was done in previous studies (by the authors) and what is new in this study. In addition, it should include better definitions.

Response to R3 – Comment 6

We have modified the Introduction to address this comment. Please see our **Response to R3 – Comment 1**.

R3 – Comment 7

On the other side **the reset of the paper is a bit long** and if possible, I would recommend shorting it I think the authors repeat sentences. I think the paper should be re-written, the introduction should provide a better discretion of where we are heading and the rest should be shorter.

Response to R3 – Comment 7

We have now gone through the paper to remove redundant information and repetitive sentences. This especially applies to the Results section where we have focused on presented the main findings of the paper. As for rewriting the Introduction, we have also done that with the goal of keeping it concise and clear about the direction of the paper and how it compares to previous work (see **Response to R3 – Comment 1**).

R3 – Comment 8

I think they might be able to combine Figure 3 & 4 to one figure.

Response to R3 – Comment 8

Figure 3 allows readers to better appreciate the discrepancy in the reliability of AI-based explanations across surgeon sub-cohorts, and to identify the particular surgeon sub-cohort (e.g., novice) that would be disadvantaged by the AI system.

In contrast, Figure 4 demonstrates the benefit of TWIX in mitigating the explanation bias presented in Figure 3. As such, we believe that combining Figure 3 and Figure 4 into one single figure may obfuscate this important distinction while overwhelming the reader with too much information at once.

R3 – Comment 9

I find it very surprising that in figure 2, USC has lower results considering the fact the model was trained using USC.

Response to R3 – Comment 9

We hypothesize that this finding is due to the higher degree of variability in surgical activity depicted in the USC videos relative to the videos from the other hospitals. This variability might be driven by the larger number of novice surgeons (trainees) who can exhibit a wider range of surgical activity than expert surgeons. We have now included this description in the **Results → Reliability of explanations is inconsistent across hospitals (page 3)**.

We hypothesize that this finding is due to the higher degree of variability in surgical activity depicted in the USC videos relative to the videos from the other hospitals. This variability might be driven by the larger number of novice surgeons who can exhibit a wider range of surgical activity than expert surgeons.

REVIEWERS' COMMENTS:

Reviewer #1 (Remarks to the Author):

I would like to thank the authors for the carefully considered replies and also the revised manuscript. This is a sound contribution to an important field to expand.

Reviewer #3 (Remarks to the Author):

In this updated manuscript, the authors address all my previous comments. I recommend accepting this paper.